# A Laplace-inspired Distribution on SO(3) for Probabilistic Rotation Estimation

**Yingda Yin**     **Yang Wang**     **He Wang**[†]     **Baoquan Chen**[†]
Peking University

## Abstract

Estimating the 3DoF rotation from a single RGB image is an important yet challenging problem. Probabilistic rotation regression has raised more and more attention with the benefit of expressing uncertainty information along with the prediction. Though modeling noise using Gaussian-resembling Bingham distribution and matrix Fisher distribution is natural, they are shown to be sensitive to outliers for the nature of quadratic punishment to deviations. In this paper, we draw inspiration from multivariate Laplace distribution and propose a novel Rotation Laplace distribution on SO(3). Rotation Laplace distribution is robust to the disturbance of outliers and enforces much gradient to the low-error region, resulting in a better convergence. Our extensive experiments show that our proposed distribution achieves state-of-the-art performance for rotation regression tasks over both probabilistic and non-probabilistic baselines. Our project page is at pku-epic.github.io/RotationLaplace.

## 1 Introduction

Incorporating neural networks to perform rotation regression is of great importance in the field of computer vision, computer graphics and robotics (Wang et al., 2019b; Yin et al., 2022; Dong et al., 2021; Breyer et al., 2021). To close the gap between the SO(3) manifold and the Euclidean space where neural network outputs exist, one popular line of research discovers learning-friendly rotation representations including 6D continuous representation (Zhou et al., 2019), 9D matrix representation with SVD orthogonalization (Levinson et al., 2020), etc. Recently, Chen et al. (2022) focuses on the gradient backpropagating process and replaces the vanilla auto differentiation with a SO(3) manifold-aware gradient layer, which sets the new state-of-the-art in rotation regression tasks.

Reasoning about the uncertainty information along with the predicted rotation is also attracting more and more attention, which enables many applications in aerospace (Crassidis & Markley, 2003), autonomous driving (McAllister et al., 2017) and localization (Fang et al., 2020). On this front, recent efforts have been developed to model the uncertainty of rotation regression via probabilistic modeling of rotation space. The most commonly used distributions are Bingham distribution (Bingham, 1974) on $\mathcal{S}^3$ for unit quaternions and matrix Fisher distribution (Khatri & Mardia, 1977) on SO(3) for rotation matrices. These two distributions are equivalent to each other (Prentice, 1986) and resemble the Gaussian distribution in Euclidean Space (Bingham, 1974; Khatri & Mardia, 1977). While modeling noise using Gaussian-like distributions is well-motivated by the Central Limit Theorem, Gaussian distribution is well-known to be sensitive to outliers in the probabilistic regression models (Murphy, 2012). This is because Gaussian distribution penalizes deviations quadratically, so predictions with larger errors weigh much more heavily with the learning than low-error ones and thus potentially result in suboptimal convergence when a certain amount of outliers exhibit.

Unfortunately, in certain rotation regression tasks, we fairly often come across large prediction errors, *e.g.* 180° error, due to either the (near) symmetry nature of the objects or severe occlusions (Murphy et al., 2021). In Fig. 1(left), using training on single image rotation regression as an example, we show the statistics of predictions after achieving convergence, assuming matrix Fisher distribution (as done in Mohlin et al. (2020)). The blue histogram shows the population with different prediction errors and the red dots are the impacts of these predictions on learning, evaluated

---

[†]He Wang and Baoquan Chen are the corresponding authors ({hewang, baoquan}@pku.edu.cn).

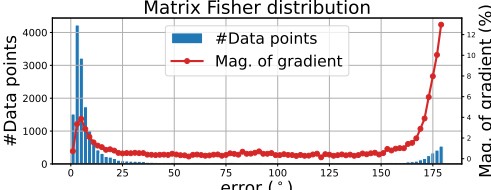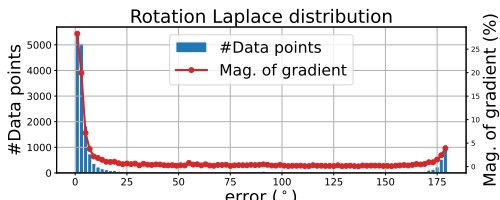

Figure 1: **Visualization of the results of matrix Fisher distribution and Rotation Laplace distribution after convergence.** The horizontal axis is the geodesic distance between the prediction and the ground truth. The blue bins count the number of data points within corresponding errors (2° each bin). The red dots illustrate the percentage of the sum of the gradient magnitude $\|\partial\mathcal{L}/\partial(\text{dist. param.})\|$ within each bin. The experiment is done on all categories of ModelNet10-SO3 dataset.

by computing the sum of their gradient magnitudes $\|\partial\mathcal{L}/\partial(\text{distribution param.})\|$ within each bin and then normalizing them across bins. It is clear that the 180° outliers dominate the gradient as well as the network training though their population is tiny, while the vast majority of points with low error predictions are deprioritized. Arguably, at convergence, the gradient should focus more on refining the low errors rather than fixing the inevitable large errors (*e.g.* arose from symmetry). This motivates us to find a better probabilistic model for rotation.

As pointed out by Murphy (2012), Laplace distribution, with heavy tails, is a better option for robust probabilistic modeling. Laplace distribution drops sharply around its mode and thus allocates most of its probability density to a small region around the mode; meanwhile, it also tolerates and assigns higher likelihoods to the outliers, compared to Gaussian distribution. Consequently, it encourages predictions near its mode to be even closer, thus fitting *sparse* data well, most of whose data points are close to their mean with the exception of several outliers(Mitianoudis, 2012), which makes Laplace distribution to be favored in the context of deep learning(Goodfellow et al., 2016).

In this work, we propose a novel Laplace-inspired distribution on $\mathrm{SO}(3)$ for rotation matrices, namely Rotation Laplace distribution, for probabilistic rotation regression. We devise Rotation Laplace distribution to be an approximation of multivariate Laplace distribution in the tangent space of its mode. As shown in the visualization in Fig. 1(right), our Rotation Laplace distribution is robust to the disturbance of outliers, with most of its gradient contributed by the low-error region, and thus leads to a better convergence along with significantly higher accuracy. Moreover, our Rotation Laplace distribution is simply parameterized by an unconstrained $3 \times 3$ matrix and thus accommodates the Euclidean output of neural networks with ease. This network-friendly distribution requires neither complex functions to fulfill the constraints of parameterization nor any normalization process from Euclidean to rotation manifold which has been shown harmful for learning (Chen et al., 2022).

For completeness of the derivations, we also propose the Laplace-inspired distribution on $\mathcal{S}^3$ for quaternions. We show that Rotation Laplace distribution is equivalent to Quaternion Laplace distribution, similar to the equivalence of matrix Fisher distribution and Bingham distribution.

We extensively compare our Rotation Laplace distributions to methods that parameterize distributions on $\mathrm{SO}(3)$ for pose estimation, and also non-probabilistic approaches including multiple rotation representations and recent $\mathrm{SO}(3)$-aware gradient layer (Chen et al., 2022). On common benchmark datasets of rotation estimation from RGB images, we achieve a significant and consistent performance improvement over all baselines.

## 2 RELATED WORK

**Probabilistic regression**  Nix & Weigend (1994) first proposes to model the output of the neural network as a Gaussian distribution and learn the Gaussian parameters by the negative log-likelihood loss function, through which one obtains not only the target but also a measure of prediction uncertainty. More recently, Kendall & Gal (2017) offers more understanding and analysis of the underlying uncertainties. Lakshminarayanan et al. (2017) further improves the performance of uncertainty estimation by network ensembling and adversarial training. Makansi et al. (2019) stabilizes the training with the winner-takes-all and iterative grouping strategies. Probabilistic regression for

uncertainty prediction has been widely used in various applications, including optical flow estimation(Ilg et al., 2018), depth estimation (Poggi et al., 2020), weather forecasting (Wang et al., 2019a), *etc.*

Among the literature of decades, the majority of probabilistic regression works model the network output by a Gaussian-like distribution, while Laplace distribution is less discovered. Li et al. (2021) empirically finds that assuming a Laplace distribution in the process of maximum likelihood estimation yields better performance than a Gaussian distribution, in the field of 3D human pose estimation. Recent work (Nair et al., 2022) makes use of Laplace distribution to improve the robustness of maximum likelihood-based uncertainty estimation. Due to the heavy-tailed property of Laplace distribution, the outlier data produces comparatively less loss and have an insubstantial impact on training. Other than in Euclidean space, Mitianoudis (2012) develops Generalized Directional Laplacian distribution in $\mathcal{S}^d$ for the application of audio separation.

**Probabilistic rotation regression** Several works focus on utilizing probability distributions on the rotation manifold for rotation uncertainty estimation. Prokudin et al. (2018) uses the mixture of von Mises distributions (Mardia et al., 2000) over Euler angles using Biternion networks. In Gilitschenski et al. (2019) and Deng et al. (2022), Bingham distribution over unit quaternion is used to jointly estimate a probability distribution over all axes. Mohlin et al. (2020) leverages matrix Fisher distribution (Khatri & Mardia, 1977) on SO(3) over rotation matrices for deep rotation regression. Though both bear similar properties with Gaussian distribution in Euclidean space, matrix Fisher distribution benefits from the continuous rotation representation and unconstrained distribution parameters, which yields better performance (Murphy et al., 2021). Recently, Murphy et al. (2021) introduces a non-parametric implicit pdf over SO(3), with the distribution properties modeled by the neural network parameters. Implicit-pdf especially does good for modeling rotations of symmetric objects.

**Non-probabilistic rotation regression** The choice of rotation representation is one of the core issues concerning rotation regression. The commonly used representations include Euler angles (Kundu et al., 2018; Tulsiani & Malik, 2015), unit quaternion (Kendall & Cipolla, 2017; Kendall et al., 2015; Xiang et al., 2017) and axis-angle (Do et al., 2018; Gao et al., 2018; Ummenhofer et al., 2017), *etc.* However, Euler angles may suffer from gimbal lock, and unit quaternions doubly cover the group of SO(3), which leads to two disconnected local minima. Moreover, Zhou et al. (2019) points out that all representations in the real Euclidean spaces of four or fewer dimensions are discontinuous and are not friendly for deep learning. To this end, the continuous 6D representation with Gram-Schmidt orthogonalization (Zhou et al., 2019) and 9D representation with SVD orthogonalization (Levinson et al., 2020) have been proposed, respectively. More recently, Chen et al. (2022) investigates the gradient backpropagation in the backward pass and proposes a SO(3) manifold-aware gradient layer.

## 3 REVISIT MATRIX FISHER DISTRIBUTION

### 3.1 MATRIX FISHER DISTRIBUTION

Matrix Fisher distribution (or von Mises-Fisher matrix distribution) (Khatri & Mardia, 1977) is one of the widely used distributions for probabilistic modeling of rotation matrices.

**Definition 1. Matrix Fisher distribution**. *The random variable* $\mathbf{R} \in \mathrm{SO}(3)$ *follows matrix Fisher distribution with parameter* $\mathbf{A}$*, if its probability density function is defined as*

$$p(\mathbf{R}; \mathbf{A}) = \frac{1}{F(\mathbf{A})} \exp\left(\mathrm{tr}(\mathbf{A}^T \mathbf{R})\right) \tag{1}$$

*where* $\mathbf{A} \in \mathbb{R}^{3\times3}$ *is an unconstrained matrix, and* $F(\mathbf{A}) \in \mathbb{R}$ *is the normalization factor. Without further clarification, we denote* $F$ *as the normalization factor of the corresponding distribution in the remaining of this paper. We also denote matrix Fisher distribution as* $\mathbf{R} \sim \mathcal{MF}(\mathbf{A})$*.*

Suppose the singular value decomposition of matrix $\mathbf{A}$ is given by $\mathbf{A} = \mathbf{U}'\mathbf{S}'(\mathbf{V}')^T$, *proper* SVD is defined as $\mathbf{A} = \mathbf{U}\mathbf{S}\mathbf{V}^T$ where

$$\mathbf{U} = \mathbf{U}' \operatorname{diag}(1, 1, \det(\mathbf{U}')) \qquad \mathbf{V} = \mathbf{V}' \operatorname{diag}(1, 1, \det(\mathbf{V}'))$$

$$\mathbf{S} = \operatorname{diag}(s_1, s_2, s_3) = \operatorname{diag}(s_1', s_2', \det(\mathbf{U}'\mathbf{V}')s_3')$$

The definition of $\mathbf{U}$ and $\mathbf{V}$ ensures that $\det(\mathbf{U}) = \det(\mathbf{V}) = 1$ and $\mathbf{U}, \mathbf{V} \in \mathrm{SO}(3)$.

## 3.2 Relationship between Matrix Fisher Distribution in $\mathrm{SO}(3)$ and Gaussian Distribution in $\mathbb{R}^3$

It is shown that matrix Fisher distribution is highly relevant with zero-mean Gaussian distribution near its mode (Lee, 2018a;b). Denote $\mathbf{R}_0$ as the mode of matrix Fisher distribution, and define $\widetilde{\mathbf{R}} = \mathbf{R}_0^T \mathbf{R}$, the relationship is shown as follows. Please refer to supplementary for the proof.

**Proposition 1.** *Let* $\mathbf{\Phi} = \log \widetilde{\mathbf{R}} \in \mathfrak{so}(3)$ *and* $\phi = \mathbf{\Phi}^\vee \in \mathbb{R}^3$. *For rotation matrix* $\mathbf{R} \in \mathrm{SO}(3)$ *following* matrix Fisher distribution, *when* $\|\mathbf{R} - \mathbf{R}_0\| \to 0$ , $\phi$ *follows zero-mean* multivariate Gaussian distribution.

# 4 Probabilistic Rotation Estimation with Rotation Laplace Distribution

## 4.1 Rotation Laplace Distribution

We get inspiration from multivariate Laplace distribution (Eltoft et al., 2006; Kozubowski et al., 2013), defined as follows.

**Definition 2. Multivariate Laplace distribution.** *If means* $\boldsymbol{\mu} = \mathbf{0}$, *the d-dimensional multivariate Laplace distribution with covariance matrix* $\mathbf{\Sigma}$ *is defined as*

$$p(\mathbf{x}; \mathbf{\Sigma}) = \frac{1}{F} \left( \mathbf{x}^T \mathbf{\Sigma}^{-1} \mathbf{x} \right)^{v/2} K_v \left( \sqrt{2\mathbf{x}^T \mathbf{\Sigma}^{-1} \mathbf{x}} \right)$$

*where* $v = (2 - d)/2$ *and* $K_v$ *is modified Bessel function of the second kind.*

We consider three dimensional Laplace distribution of $\mathbf{x} \in \mathbb{R}^3$ (i.e. $d = 3$ and $v = -\frac{1}{2}$). Given the property $K_{-\frac{1}{2}}(\xi) \propto \xi^{-\frac{1}{2}} \exp(-\xi)$, three dimensional Laplace distribution is defined as

$$p(\mathbf{x}; \mathbf{\Sigma}) = \frac{1}{F} \frac{\exp\left(-\sqrt{2\mathbf{x}^T \mathbf{\Sigma}^{-1} \mathbf{x}}\right)}{\sqrt{\mathbf{x}^T \mathbf{\Sigma}^{-1} \mathbf{x}}}$$

In this section, we first give the definition of our proposed Rotation Laplace distribution and then shows its relationship with multivariate Laplace distribution.

**Definition 3. Rotation Laplace distribution.** *The random variable* $\mathbf{R} \in \mathrm{SO}(3)$ *follows Rotation Laplace distribution with parameter* $\mathbf{A}$, *if its probability density function is defined as*

$$p(\mathbf{R}; \mathbf{A}) = \frac{1}{F(\mathbf{A})} \frac{\exp\left(-\sqrt{\mathrm{tr}\left(\mathbf{S} - \mathbf{A}^T \mathbf{R}\right)}\right)}{\sqrt{\mathrm{tr}\left(\mathbf{S} - \mathbf{A}^T \mathbf{R}\right)}} \tag{2}$$

*where* $\mathbf{A} \in \mathbb{R}^{3 \times 3}$ *is an unconstrained matrix, and* $\mathbf{S}$ *is the diagonal matrix composed of the proper singular values of matrix* $\mathbf{A}$, *i.e.,* $\mathbf{A} = \mathbf{U}\mathbf{S}\mathbf{V}^T$. *We also denote Rotation Laplace distribution as* $\mathbf{R} \sim \mathcal{RL}(\mathbf{A})$.

Denote $\mathbf{R}_0$ as the mode of Rotation Laplace distribution and define $\widetilde{\mathbf{R}} = \mathbf{R}_0^T \mathbf{R}$, the relationship between Rotation Laplace distribution and multivariate Laplace distribution is shown as follows.

**Proposition 2.** *Let* $\mathbf{\Phi} = \log \widetilde{\mathbf{R}} \in \mathfrak{so}(3)$ *and* $\phi = \mathbf{\Phi}^\vee \in \mathbb{R}^3$. *For rotation matrix* $\mathbf{R} \in \mathrm{SO}(3)$ *following* Rotation Laplace distribution, *when* $\|\mathbf{R} - \mathbf{R}_0\| \to 0$ , $\phi$ *follows zero-mean* multivariate Laplace distribution.

*Proof.* Apply proper SVD to matrix $\mathbf{A}$ as $\mathbf{A} = \mathbf{U}\mathbf{S}\mathbf{V}^T$. For $\mathbf{R} \sim \mathcal{RL}(\mathbf{A})$ , we have

$$p(\mathbf{R})\mathrm{d}\mathbf{R} \propto \frac{\exp\left(\sqrt{\mathrm{tr}(\mathbf{S} - \mathbf{A}^T \mathbf{R})}\right)}{\sqrt{\mathrm{tr}(\mathbf{S} - \mathbf{A}^T \mathbf{R})}} \mathrm{d}\mathbf{R} = \frac{\exp\left(\sqrt{\mathrm{tr}(\mathbf{S} - \mathbf{S}\mathbf{V}^T \widetilde{\mathbf{R}} \mathbf{V})}\right)}{\sqrt{\mathrm{tr}(\mathbf{S} - \mathbf{S}\mathbf{V}^T \widetilde{\mathbf{R}} \mathbf{V})}} \mathrm{d}\mathbf{R} \tag{3}$$

With $\phi = (\log \widetilde{\mathbf{R}})^\vee \in \mathbb{R}^3$, $\widetilde{\mathbf{R}}$ can be parameterized as

$$\widetilde{\mathbf{R}}(\phi) = \exp(\hat{\phi}) = \mathbf{I} + \frac{\sin \|\phi\|}{\|\phi\|}\hat{\phi} + \frac{1 - \cos \|\phi\|}{\|\phi\|^2}\hat{\phi}^2$$

We follow the common practice (Mohlin et al., 2020; Lee, 2018a) that the Haar measure $\mathrm{d}\mathbf{R}$ is scaled such that $\int_{SO(3)} \mathrm{d}\mathbf{R} = 1$ and thus the Haar measure is given by

$$\mathrm{d}\widetilde{\mathbf{R}} = \frac{1 - \cos \|\phi\|}{4\pi^2 \|\phi\|^2}\mathrm{d}\phi = \left( \frac{1}{8\pi^2} + O(\|\phi\|)^2 \right)\mathrm{d}\phi. \tag{4}$$

Also, $\widetilde{\mathbf{R}}$ expanded at $\phi = \mathbf{0}$ is computed as $\widetilde{\mathbf{R}} = \mathbf{I} + \hat{\phi} + \frac{1}{2}\hat{\phi}^2 + O(\|\phi\|^3)$, we have

$$\mathbf{V}^T \widetilde{\mathbf{R}} \mathbf{V} = \mathbf{I} + \mathbf{V}^T \hat{\phi} \mathbf{V} + \frac{1}{2}\mathbf{V}^T \hat{\phi}^2 \mathbf{V} + O(\|\phi\|^3) = \mathbf{I} + \widehat{\mathbf{V}^T \phi} + \frac{1}{2}\widehat{\mathbf{V}^T \phi}^2 + O(\|\phi\|^3)$$

$$= \begin{bmatrix} 1 - \frac{1}{2}(\mu_2^2 + \mu_3^2) & \frac{1}{2}\mu_1\mu_2 - \mu_3 & \frac{1}{2}\mu_1\mu_3 + \mu_2 \\ \frac{1}{2}\mu_1\mu_2 + \mu_3 & 1 - \frac{1}{2}(\mu_3^2 + \mu_1^2) & \frac{1}{2}\mu_2\mu_3 - \mu_1 \\ \frac{1}{2}\mu_1\mu_3 - \mu_2 & \frac{1}{2}\mu_2\mu_3 + \mu_1 & 1 - \frac{1}{2}(\mu_1^2 + \mu_2^2) \end{bmatrix} + O(\|\phi\|^3), \tag{5}$$

where $(\mu_1, \mu_2, \mu_3)^T = \mathbf{V}^T \phi$, and

$$\mathrm{tr}(\mathbf{S}\text{-}\mathbf{S}\mathbf{V}^T \widetilde{\mathbf{R}} \mathbf{V}) = \sum_{(i,j,k) \in I} \frac{1}{2}(s_j + s_k)\mu_i^2 + O(\|\phi\|^3) = \frac{1}{2}\phi^T \mathbf{V} \begin{bmatrix} s_2 + s_3 & & \\ & s_1 + s_3 & \\ & & s_1 + s_2 \end{bmatrix}\mathbf{V}^T \phi + O(\|\phi\|^3) \tag{6}$$

Considering Eq. 3, 4 and 6, we have

$$p(\mathbf{R})\mathrm{d}\mathbf{R} \propto \frac{\exp\left(\sqrt{\mathrm{tr}(\mathbf{S}\text{-}\mathbf{A}^T\mathbf{R})}\right)}{\sqrt{\mathrm{tr}(\mathbf{S}\text{-}\mathbf{A}^T\mathbf{R})}}\mathrm{d}\mathbf{R} = \frac{1}{8\pi^2}\frac{\exp\left(-\sqrt{2\phi^T \Sigma^{-1}\phi}\right)}{\sqrt{2\phi^T \Sigma^{-1}\phi}}\left(1 + O(\|\phi\|^2)\right)\mathrm{d}\phi \tag{7}$$

When $\|\mathbf{R} - \mathbf{R}_0\| \to 0$, we have $\|\widetilde{\mathbf{R}} - \mathbf{I}\| \to 0$ and $\phi \to \mathbf{0}$, so Eq. 7 follows the multivariate Laplace distribution with the covariance matrix as $\Sigma$, where $\Sigma = 4\mathbf{V}\,\mathrm{diag}(\frac{1}{s_2+s_3}, \frac{1}{s_1+s_3}, \frac{1}{s_1+s_2})\mathbf{V}^T$. $\square$

Rotation Laplace distribution bears similar properties with matrix Fisher distribution. Its mode is computed as $\mathbf{U}\mathbf{V}^T$. The columns of $\mathbf{U}$ and the proper singular values $\mathbf{S}$ describe the orientation and the strength of dispersions, respectively.

## 4.2 NEGATIVE LOG-LIKELIHOOD LOSS

Given a collection of observations $\mathcal{X} = \{\boldsymbol{x}_i\}$ and the associated ground truth rotations $\mathcal{R} = \{\mathbf{R}_i\}$, we aim at training the network to best estimate the parameter $\mathbf{A}$ of Rotation Laplace distribution. This is achieved by maximizing a likelihood function so that, under our probabilistic model, the observed data is most probable, which is known as maximum likelihood estimation (MLE). We use the negative log-likelihood of $\mathbf{R}_{\boldsymbol{x}}$ as the loss function:

$$\mathcal{L}(\boldsymbol{x}, \mathbf{R}_{\boldsymbol{x}}) = -\log p\left(\mathbf{R}_{\boldsymbol{x}}; \mathbf{A}_{\boldsymbol{x}}\right)$$

## 4.3 DISCRETE APPROXIMATION OF THE NORMALIZATION FACTOR

Efficiently and accurately estimating the normalization factor for distributions over $SO(3)$ is nontrivial. Inspired by Murphy et al. (2021), we approximate the normalization factor of Rotation Laplace distribution through equivolumetric discretization over $SO(3)$ manifold. We employ the discretization method introduced in Yershova et al. (2010), which starts with the equal area grids on the 2-sphere (Gorski et al., 2005) and covers $SO(3)$ by threading a great circle through each point on the surface of a 2-sphere with Hopf fibration. Concretely, we discretize $SO(3)$ space into a finite set of equivolumetric grids $\mathcal{G} = \{\mathbf{R} | \mathbf{R} \in SO(3)\}$, the normalization factor of Laplace Rotation distribution is computed as

$$F(\mathbf{A}) = \int_{SO(3)} \frac{\exp\left(-\sqrt{\mathrm{tr}\left(\mathbf{S} - \mathbf{A}^T \mathbf{R}\right)}\right)}{\sqrt{\mathrm{tr}\left(\mathbf{S} - \mathbf{A}^T \mathbf{R}\right)}}\mathrm{d}\mathbf{R} \approx \sum_{\mathbf{R}_i \in \mathcal{G}} \frac{\exp\left(-\sqrt{\mathrm{tr}\left(\mathbf{S} - \mathbf{A}^T \mathbf{R}_i\right)}\right)}{\sqrt{\mathrm{tr}\left(\mathbf{S} - \mathbf{A}^T \mathbf{R}_i\right)}}\Delta\mathbf{R}_i$$

where $\Delta\mathbf{R}_i = \frac{\int_{SO(3)} \mathrm{d}\mathbf{R}}{|\mathcal{G}|} = \frac{1}{|\mathcal{G}|}$. In experiments, we discretize $SO(3)$ space into about 37k points. Please refer to supplementary for analysis of the effect of different numbers of samples.

### 4.4 QUATERNION LAPLACE DISTRIBUTION

In this section, we introduce our extension of Laplace-inspired distribution for quaternions, namely, Quaternion Laplace distribution.

**Definition 4. Quaternion Laplace distribution.** *The random variable* $\mathbf{q} \in \mathcal{S}^3$ *follows Quaternion Laplace distribution with parameter* $\mathbf{M}$ *and* $\mathbf{Z}$, *if its probability density function is defined as*

$$p(\mathbf{q}; \mathbf{M}, \mathbf{Z}) = \frac{1}{F(\mathbf{Z})} \frac{\exp\left(-\sqrt{-\mathbf{q}^T \mathbf{M} \mathbf{Z} \mathbf{M}^T \mathbf{q}}\right)}{\sqrt{-\mathbf{q}^T \mathbf{M} \mathbf{Z} \mathbf{M}^T \mathbf{q}}} \tag{8}$$

*where* $\mathbf{M} \in \mathbf{O}(4)$ *is a* $4 \times 4$ *orthogonal matrix, and* $\mathbf{Z} = \mathrm{diag}(0, z_1, z_2, z_3)$ *is a* $4 \times 4$ *diagonal matrix with* $0 \geq z_1 \geq z_2 \geq z_3$. *We also denote Quaternion Laplace distribution as* $\mathbf{q} \sim \mathcal{QL}(\mathbf{M}, \mathbf{Z})$.

**Proposition 3.** *Denote* $\mathbf{q}_0$ *as the mode of Quaternion Laplace distribution. Let* $\pi$ *be the tangent space of* $\mathbb{S}^3$ *at* $\mathbf{q}_0$, *and* $\pi(\mathbf{x}) \in \mathbb{R}^4$ *be the projection of* $\mathbf{x} \in \mathbb{R}^4$ *on* $\pi$. *For quaternion* $\mathbf{q} \in \mathbb{S}^3$ *following* Bingham distribution / Quaternion Laplace distribution, *when* $\mathbf{q} \to \mathbf{q}_0$, $\pi(\mathbf{q})$ *follows zero-mean* multivariate Gaussian distribution / *zero-mean* multivariate Laplace distribution.

Both Bingham distribution and Quaternion Laplace distribution exhibit antipodal symmetry on $\mathcal{S}^3$, i.e., $p(\mathbf{q}) = p(-\mathbf{q})$, which captures the nature that the quaternions $\mathbf{q}$ and $-\mathbf{q}$ represent the same rotation on $\mathrm{SO}(3)$.

**Proposition 4.** *Denote* $\gamma$ *as the standard transformation from unit quaternions to corresponding rotation matrices. For rotation matrix* $\mathbf{R} \in \mathrm{SO}(3)$ *following* Rotation Laplace distribution, $\mathbf{q} = \gamma^{-1}(\mathbf{R}) \in \mathbb{S}^3$ *follows* Quaternion Laplace distribution.

Prop. 4 shows that our proposed Rotation Laplace distribution is equivalent to Quaternion Laplace distribution, similar to the equivalence of matrix Fisher distribution and Bingham distribution (Prentice, 1986), demonstrating the consistency of our derivations. Please see supplementary for the proofs to the above propositions.

The normalization factor of Quaternion Laplace distribution is also approximated by dense discretization, as follows:

$$F(\mathbf{Z}) = \oint_{\mathcal{S}^3} \frac{\exp\left(-\sqrt{-\mathbf{q}^T \mathbf{M} \mathbf{Z} \mathbf{M}^T \mathbf{q}}\right)}{\sqrt{-\mathbf{q}^T \mathbf{M} \mathbf{Z} \mathbf{M}^T \mathbf{q}}} \mathrm{d}\mathbf{q} \approx \sum_{\mathbf{q}_i \in \mathcal{G}_{\mathbf{q}}} \frac{\exp\left(-\sqrt{-\mathbf{q}_i^T \mathbf{M} \mathbf{Z} \mathbf{M}^T \mathbf{q}_i}\right)}{\sqrt{-\mathbf{q}_i^T \mathbf{M} \mathbf{Z} \mathbf{M}^T \mathbf{q}_i}} \Delta\mathbf{q}_i$$

where $\mathcal{G}_{\mathbf{q}} = \left\{\mathbf{q} | \mathbf{q} \in \mathcal{S}^3\right\}$ denotes the set of equivolumetric grids and $\Delta\mathbf{q}_i = \frac{\oint_{\mathcal{S}^3} \mathrm{d}\mathbf{q}}{|\mathcal{G}_{\mathbf{q}}|} = \frac{2\pi^2}{|\mathcal{G}_{\mathbf{q}}|}$.

## 5 EXPERIMENT

Following the previous state-of-the-arts (Murphy et al., 2021; Mohlin et al., 2020), we evaluate our method on the task of object rotation estimation from single RGB images, where object rotation is the relative rotation between the input object and the object in the canonical pose. Concerning this task, we find two kinds of independent research tracks with slightly different evaluation settings. One line of research focuses on probabilistic rotation regression with different parametric or non-parametric distributions on $\mathrm{SO}(3)$ (Prokudin et al., 2018; Gilitschenski et al., 2019; Deng et al., 2022; Mohlin et al., 2020; Murphy et al., 2021), and the other non-probabilistic track proposes multiple rotation representations (Zhou et al., 2019; Levinson et al., 2020; Peretroukhin et al., 2020) or improves the gradient of backpropagation (Chen et al., 2022). To fully demonstrate the capacity of our Rotation Laplace distribution, we leave the baselines in their original optimal states and adapt our method to follow the common experimental settings in each track, respectively.

### 5.1 DATASETS & EVALUATION METRICS

**Datasets** **ModelNet10-SO3** (Liao et al., 2019) is a commonly used synthetic dataset for single image rotation estimation containing 10 object classes. It is synthesized by rendering the CAD models of ModelNet-10 dataset (Wu et al., 2015) that are rotated by uniformly sampled rotations in $\mathrm{SO}(3)$. **Pascal3D+** (Xiang et al., 2014) is a popular benchmark on real-world images for pose estimation. It covers 12 common daily object categories. The images in Pascal3D+ dataset are sourced from Pascal VOC and ImageNet datasets, and are split into ImageNet_train, ImageNet_val, PascalVOC_train, and PascalVOC_val sets.

Table 1: Numerical comparisons with probabilistic baselines on ModelNet10-SO3 dataset averaged on all categories. Numbers in parentheses (·) are our reproduced results. Please refer to supplementary for comparisons with each category.

|  | Acc@3°↑ | Acc@5°↑ | Acc@10°↑ | Acc@15°↑ | Acc@30°↑ | Med.(°)↓ |
|---|---|---|---|---|---|---|
| Liao et al. (2019) | - | - | - | 0.496 | 0.658 | 28.7 |
| Prokudin et al. (2018) | - | - | - | 0.456 | 0.528 | 49.3 |
| Deng et al. (2022) | (0.138) | (0.301) | (0.502) | 0.562 (0.584) | 0.694 (0.673) | 32.6 (31.6) |
| Mohlin et al. (2020) | (0.164) | (0.389) | (0.615) | 0.693 (0.684) | 0.757 (0.751) | 17.1 (17.9) |
| Murphy et al. (2021) | (0.294) | (0.534) | (0.680) | 0.719 (0.714) | 0.735 (0.730) | 21.5 (20.3) |
| Rotation Laplace | **0.447** | **0.611** | **0.715** | **0.742** | **0.772** | **12.7** |

Table 2: Numerical comparisons with probabilistic baselines on Pascal3D+ dataset averaged on all categories. Numbers in parentheses (·) are our reproduced results. Please refer to supplementary for comparisons with each category.

|  | Acc@3°↑ | Acc@5°↑ | Acc@10°↑ | Acc@15°↑ | Acc@30°↑ | Med.(°)↓ |
|---|---|---|---|---|---|---|
| Tulsiani & Malik (2015) | - | - | - | - | 0.808 | 13.6 |
| Mahendran et al. (2018) | - | - | - | - | 0.859 | 10.1 |
| Liao et al. (2019) | - | - | - | - | 0.819 | 13.0 |
| Prokudin et al. (2018) | - | - | - | - | 0.838 | 12.2 |
| Mohlin et al. (2020) | (0.089) | (0.215) | (0.484) | (0.650) | 0.825 (0.827) | 11.5 (11.9) |
| Murphy et al. (2021) | (0.102) | (0.242) | (0.524) | (0.672) | 0.837 (0.838) | 10.3 (10.2) |
| Rotation Laplace | **0.134** | **0.292** | **0.574** | **0.714** | **0.874** | **9.3** |

**Evaluation metrics** We evaluate our experiments with the geodesic distance of the network prediction and the ground truth. This metric returns the angular error and we measure it in degrees. In addition, we report the prediction accuracy within the given error threshold.

## 5.2 Comparisons with Probabilistic Methods

### 5.2.1 Evaluation Setup

**Settings** In this section, we follow the experiment settings of the latest work (Murphy et al., 2021) and quote its reported numbers for baselines. Specifically, we train one single model for all categories of each dataset. For Pascal3D+ dataset, we follow Murphy et al. (2021) to use (the more challenging) PascalVOC_val as test set. Note that Murphy et al. (2021) only measure the coarse-scale accuracy (e.g., Acc@30°) which may not adequately satisfy the downstream tasks (Wang et al., 2019b; Fang et al., 2020). To facilitate finer-scale comparisons (e.g., Acc@5°), we further re-run several recent baselines and report the reproduced results in parentheses (·).

**Baselines** We compare our method to recent works which utilize probabilistic distributions on SO(3) for the purpose of pose estimation. In concrete, the baselines are with mixture of *von Mises* distributions **Prokudin et al. (2018)**, *Bingham* distribution **Gilitschenski et al. (2019); Deng et al. (2022)**, *matrix Fisher* distribution **Mohlin et al. (2020)** and Implicit-PDF **Murphy et al. (2021)**. We also compare to the spherical regression work of **Liao et al. (2019)** as Murphy et al. (2021) does.

### 5.2.2 Results

Table 1 shows the quantitative comparisons of our method and baselines on ModelNet10-SO3 dataset. From the multiple evaluation metrics, we can see that maximum likelihood estimation with the assumption of Rotation Laplace distribution significantly outperforms the other distributions for rotation, including matrix Fisher distribution (Mohlin et al., 2020), Bingham distribution (Do et al., 2018) and von-Mises distribution (Prokudin et al., 2018). Our method also gets superior performance than the non-parametric implicit-PDF (Murphy et al., 2021). Especially, our method improves the fine-scale Acc@3° and Acc@5° accuracy by a large margin, showing its capacity to precisely model the target distribution.

The experiments on Pascal3D+ dataset are shown in Table 2, where our Rotation Laplace distribution outperforms all the baselines. While our method gets reasonably good performance on the median

Table 3: Numerical comparisons with non-probabilistic baselines on ModelNet10-SO3 dataset. One model is trained for each category.

| Methods | Chair | | | Sofa | | | Toilet | | | Bed | | |
|---|---|---|---|---|---|---|---|---|---|---|---|---|
| | Mean↓ | Med.↓ | Acc@5↑ | Mean↓ | Med.↓ | Acc@5↑ | Mean↓ | Med.↓ | Acc@5↑ | Mean↓ | Med.↓ | Acc@5↑ |
| 6D | 19.6 | 9.1 | 0.19 | 17.5 | 7.3 | 0.27 | 10.9 | 6.2 | 0.37 | 32.3 | 11.7 | 0.11 |
| 9D | 17.5 | 8.3 | 0.23 | 19.8 | 7.6 | 0.25 | 11.8 | 6.5 | 0.34 | 30.4 | 11.1 | 0.13 |
| 9D-Inf | 12.1 | 5.1 | 0.49 | 12.5 | 3.5 | 0.70 | 7.6 | 3.7 | 0.67 | 22.5 | 4.5 | 0.56 |
| 10D | 18.4 | 9.0 | 0.20 | 20.9 | 8.7 | 0.20 | 11.5 | 5.9 | 0.39 | 29.9 | 11.5 | 0.11 |
| RPMG-6D | 12.9 | 4.7 | 0.53 | 11.5 | 2.8 | 0.77 | 7.8 | 3.4 | 0.71 | 20.3 | 3.6 | 0.67 |
| RPMG-9D | 11.9 | 4.4 | 0.58 | 10.5 | 2.4 | 0.82 | 7.5 | 3.2 | 0.75 | 20.0 | 2.9 | 0.76 |
| RPMG-10D | 12.8 | 4.5 | 0.55 | 11.2 | 2.4 | 0.82 | 7.2 | 3.0 | 0.76 | 19.2 | 2.9 | 0.75 |
| Rot. Laplace | **9.7** | **3.5** | **0.68** | **8.8** | **2.1** | **0.84** | **5.3** | **2.6** | **0.83** | 15.5 | **2.3** | **0.82** |

Table 4: Numerical comparisons with non-probabilistic baselines on Pascal3D+ dataset. One model is trained for each category.

| Methods | Bicycle | | | | Sofa | | | |
|---|---|---|---|---|---|---|---|---|
| | Acc@10↑ | Acc@15↑ | Acc@20↑ | Med.↓ | Acc@10↑ | Acc@15↑ | Acc@20↑ | Med.↓ |
| 6D | 0.218 | 0.390 | 0.553 | 18.1 | 0.508 | 0.767 | 0.890 | 9.9 |
| 9D | 0.206 | 0.376 | 0.569 | 18.0 | 0.524 | 0.796 | 0.903 | 9.2 |
| 9D-Inf | 0.380 | 0.533 | 0.699 | 13.4 | 0.709 | 0.880 | 0.935 | 6.7 |
| 10D | 0.239 | 0.423 | 0.567 | 17.9 | 0.502 | 0.770 | 0.896 | 9.8 |
| RPMG-6D | 0.354 | 0.572 | 0.706 | 13.5 | 0.696 | 0.861 | 0.922 | 6.7 |
| RPMG-9D | 0.368 | 0.574 | 0.718 | 12.5 | 0.725 | 0.880 | 0.958 | 6.7 |
| RPMG-10D | 0.400 | 0.577 | 0.713 | 12.9 | 0.693 | 0.871 | 0.939 | 7.0 |
| Rot. Laplace | **0.435** | **0.641** | **0.744** | **11.2** | **0.735** | **0.900** | **0.964** | **6.3** |

error and coarser-scale accuracy, we do not find a similar impressive improvement on fine-scale metrics as in ModelNet10-SO3 dataset. We suspect it is because the imperfect human annotations of real-world images may lead to comparatively noisy ground truths, increasing the difficulty for networks to get rather close predictions with GT labels. Nevertheless, our method still manages to obtain superior performance, which illustrates the robustness of our Rotation Laplace distribution.

## 5.3 COMPARISONS WITH NON-PROBABILISTIC METHODS

### 5.3.1 EVALUATION SETUP

**Settings** For comparisons with non-probabilistic methods, we follow the latest work of Chen et al. (2022) to learn a network for each category. For Pascal3D+ dataset, we follow Chen et al. (2022) to use ImageNet_val as our test set. We use the same evaluation metrics as in Chen et al. (2022) and quote its reported numbers for baselines.

**Baselines** We compare to multiple baselines that leverage different rotation representations to directly regress the prediction given input images, including **6D** (Zhou et al., 2019), **9D / 9D-Inf** (Levinson et al., 2020) and **10D** (Peretroukhin et al., 2020). We also include regularized projective manifold gradient (**RPMG**) series of methods (Chen et al., 2022).

### 5.3.2 RESULTS

We report the numerical results of our method and on-probabilistic baselines on ModelNet10-SO3 dataset in Table 3. Our method obtains a clear superior performance to the best competitor under all the metrics among all the categories. Note that we train a model for each category (so do all the baselines), thus our performance in Table 3 is better than Table 1 where one model is trained for the whole dataset. The results on Pascal3D+ dataset are shown in Table 4 where our method with Rotation Laplace distribution achieves state-of-the-art performance.

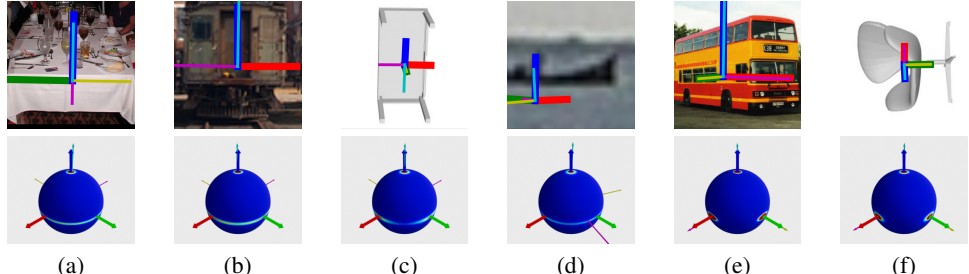

|         (a)         |         (b)         |         (c)         |         (d)         |         (e)         |         (f)         |

Figure 2: **Visualizations of the predicted distributions.** The top row displays example images with the projected axes of predictions (thick lines) and ground truths (thin lines) of the object. The bottom row shows the visualization of the corresponding predicted distributions of the image. For clarity we have aligned the predicted poses with the standard axes.

Table 5: Numerical comparisons with our proposed Quaternion & Rotation Laplace distribution and baselines on ModelNet10-SO3 dataset. One model is trained for each category. Quaternion Laplace distribution clearly outperforms Bingham distribution (Deng et al., 2022).

|  | Chair | | | Sofa | | | Toilet | | | Bed | | |
|---|---|---|---|---|---|---|---|---|---|---|---|---|
|  | Mean↓ | Med.↓ | Acc@5↑ | Mean↓ | Med.↓ | Acc@5↑ | Mean↓ | Med.↓ | Acc@5↑ | Mean↓ | Med.↓ | Acc@5↑ |
| Deng et al. (2022) | 16.5 | 7.2 | 0.31 | 16.5 | 4.9 | 0.52 | 9.6 | 4.2 | 0.59 | 22.0 | 5.1 | 0.49 |
| Mohlin et al. (2020) | 10.8 | 4.6 | 0.55 | 11.1 | 3.5 | 0.70 | 6.4 | 3.5 | 0.70 | 16.0 | 3.8 | 0.66 |
| Quat. Laplace | 12.6 | 5.2 | 0.49 | 13.1 | 3.7 | 0.67 | 5.9 | 3.4 | 0.69 | 17.7 | 3.4 | 0.69 |
| Rot. Laplace | **9.7** | **3.5** | **0.68** | **8.8** | **2.1** | **0.84** | **5.3** | **2.6** | **0.83** | **15.5** | **2.3** | **0.82** |

## 5.4 QUALITATIVE RESULTS

We visualize the predicted distributions in Figure 2 with the visualization method in Mohlin et al. (2020). As shown in the figure, the predicted distributions can exhibit high uncertainty when the object has rotational symmetry, leading to near 180° errors (a-c), or the input image is with low resolution (d). Subfigure (e-f) show cases with high certainty and reasonably low errors. Please refer to the supplementary for more visual results.

## 5.5 IMPLEMENTATION DETAILS

For fair comparisons, we follow the implementation designs of Mohlin et al. (2020) and merely change the distribution from matrix Fisher distribution to our Rotation Laplace distribution. For numerical stability, we clip $\mathrm{tr}(\mathbf{S} - \mathbf{A}^T\mathbf{R})$ by $\max(1e-8, \mathrm{tr}(\mathbf{S} - \mathbf{A}^T\mathbf{R}))$ for Eq.2. Please refer to supplementary for more details.

## 5.6 COMPARISONS OF ROTATION LAPLACE DISTRIBUTION AND QUATERNION LAPLACE DISTRIBUTION

For the completeness of experiments, we also compare our proposed Quaternion Laplace distribution and Bingham distribution and report the performance in Table 5. As shown in the table, Quaternion Laplace distribution consistently achieves superior performance than its competitor, which validates the effectiveness of our Laplace-inspired derivations. However, its rotation error is in general larger than Rotation Laplace distribution, since its rotation representation, quaternion, is not a continuous representation, as pointed in Zhou et al. (2019), thus leading to inferior performance.

## 6 CONCLUSION

In this paper, we draw inspiration from multivariant Laplace distribution and derive two novel distributions for probabilistic rotation regression, namely, Rotation Laplace distribution for rotation matrices on $\mathrm{SO}(3)$ and Quaternion Laplace distribution for quaternions on $\mathcal{S}^3$. Extensive comparisons with both probabilistic and non-probabilistic baselines on ModelNet10-SO3 and Pascal3D+ datasets demonstrate the effectiveness and advantages of our proposed distributions.

## ACKNOWLEDGEMENT

We thank Haoran Liu from Peking University for the help in experiments. This work is supported in part by National Key R&D Program of China 2022ZD0160801.

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

## A    NOTATIONS AND DEFINITIONS

### A.1    NOTATIONS FOR LIE ALGEBRA AND EXPONENTIAL & LOGARITHM MAP

This paper follows the common notations for Lie algebra and exponential & logarithm map (Lee, 2018a; Teed & Deng, 2021; Sola et al., 2018).

The three-dimensional special orthogonal group $SO(3)$ is defined as

$$SO(3) = \{\mathbf{R} \in \mathbb{R}^{3 \times 3} | \mathbf{R}\mathbf{R}^T = \mathbf{I}, \det(\mathbf{R}) = 1\}.$$

The Lie algebra of $SO(3)$, denoted by $\mathfrak{so}(3)$, is the tangent space of $SO(3)$ at $\mathbf{I}$, given by

$$\mathfrak{so}(3) = \{\mathbf{\Phi} \in \mathbb{R}^{3 \times 3} | \mathbf{\Phi} = -\mathbf{\Phi}^T\}.$$

$\mathfrak{so}(3)$ is identified with $(\mathbb{R}^3, \times)$ by the *hat* $\wedge$ map and the *vee* $\vee$ map defined as

$$\mathfrak{so}(3) \ni \begin{bmatrix} 0 & -\phi_z & \phi_y \\ \phi_z & 0 & -\phi_x \\ -\phi_y & \phi_x & 0 \end{bmatrix} \underset{\text{hat } \wedge}{\overset{\text{vee } \vee}{\rightleftarrows}} \begin{bmatrix} \phi_x \\ \phi_y \\ \phi_z \end{bmatrix} \in \mathbb{R}^3$$

The exponential map, taking skew symmetric matrices to rotation matrices is given by

$$\exp(\hat{\phi}) = \sum_{k=0}^{\infty} \frac{\hat{\phi}^k}{k!} = \mathbf{I} + \frac{\sin\theta}{\theta}\hat{\phi} + \frac{1 - \cos\theta}{\theta^2}\hat{\phi}^2,$$

where $\theta = \|\phi\|$. The exponential map can be inverted by the logarithm map, going from $SO(3)$ to $\mathfrak{so}(3)$ as

$$\log(\mathbf{R}) = \frac{\theta}{2\sin\theta}(\mathbf{R} - \mathbf{R}^T),$$

where $\theta = \arccos\frac{\operatorname{tr}(\mathbf{R})-1}{2}$.

### A.2    HAAR MEASURE

To evaluate the normalization factors and therefore the probability density functions, the measure $d\mathbf{R}$ on $SO(3)$ needs to be defined. For the Lie group $SO(3)$, the commonly used bi-invariant measure is referred to as Haar measure (Haar, 1933; James, 1999). Haar measure is unique up to scalar multiples (Chirikjian, 2000) and we follow the common practice (Mohlin et al., 2020; Lee, 2018a) that the Haar measure $d\mathbf{R}$ is scaled such that $\int_{SO(3)} d\mathbf{R} = 1$.

## B    MORE ANALYSIS ON GRADIENT W.R.T. OUTLIERS

In the task of rotation regression, predictions with really large errors (e.g., 180° error) are fairly observed due to rotational ambiguity or lack of discriminate visual features. Properly handling these outliers during training is one of the keys to success in probabilistic modeling of rotations.

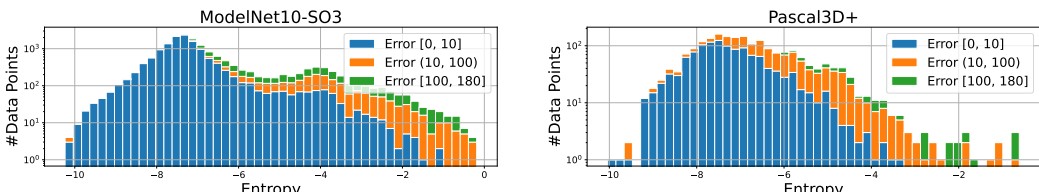

Figure 3: Visualization of the gradient magnituide $\|\partial\mathcal{L}/\partial(\text{distribution param.})\|$ w.r.t. the prediction errors on ModelNet10-SO3 dataset after convergence.

Figure 4: **Visualization of the indication ability of the distribution entropy w.r.t. the performance.** The horizontal axis is the distribution entropy and the vertical axis is the number of data points (in log scale), color coded by the errors (in degrees). The experiments are done on the test set of ModelNet10-SO3 dataset (left) and Pascal3D+ dataset (right).

In Figure 3, for matrix Fisher distribution and Rotation Laplace distribution, we visualize the gradient magnitudes $\|\partial\mathcal{L}/\partial(\text{distribution param.})\|$ w.r.t. the prediction errors on ModelNet10-SO3 dataset after convergence, where each point is a data point in the test set. As shown in the figure, for matrix Fisher distribution, predictions with larger errors clearly yield larger gradient magnitudes, and those with near 180° errors (the outliers) have the biggest impact. Given that outliers may be inevitable and hard to be fixed, they may severely disturb the training process and the sensitivity to outliers can result in a poor fit (Murphy, 2012; Nair et al., 2022). In contrast, for our Rotation Laplace distribution, the gradient magnitudes are not affected by the prediction errors much, leading to a stable learning process.

Consistent results can also be seen in Figure 1 of the main paper, where the red dots illustrate the *sum* of the gradient magnitude over the population within an interval of prediction errors. We argue that, at convergence, the gradient should focus more on the large population with low errors rather than fixing the unavoidable large errors.

## C  UNCERTAINTY QUANTIFICATION MEASURED BY DISTRIBUTION ENTROPY

Probabilistic modeling of rotation naturally models the uncertainty information of rotation regression. Yin et al. (2022) proposes to use the *entropy* of the distribution as an uncertainty measure. We adopt it as the uncertainty indicator of Rotation Laplace distribution and plot the relationship between the error of the prediction and the corresponding distribution entropy on the testset of ModelNet10-SO3 and Pascal3D+ datasets in Figure 4. As shown in the figure, predictions with lower entropies (i.e., lower uncertainty) clearly achieve higher accuracy than predictions with large entropies, demonstrating the ability of uncertainty estimation of our Rotation Laplace distribution. We compute the entropy via discretization, where SO(3) space is quantized into a finite set of equivolumetric girds $\mathcal{G} = \{\mathbf{R}|\mathbf{R} \in \text{SO}(3)\}$, and

$$H(p) = -\int_{\text{SO}(3)} p \log p \, d\mathbf{R} \approx -\sum_{\mathbf{R}_i \in \mathcal{G}} p_i \log p_i \Delta\mathbf{R}_i$$

We use about 0.3M grids to discretize SO(3) space.

## D  EFFECT OF DIFFERENT NUMBERS OF DISCRETIZATION SAMPLES

To compute the normalization factor of our distribution, we discretize SO(3) space into a finite set of equivolumetric grids using Hopf fibration. Here we show the comparison on different numbers of samples. We experiment with ModelNet10-SO3 toilet dataset on a single 3090 GPU.

Table 6: Comparison on different numbers of discretization samples. The experiment is done on ModelNet10-SO3 toilet dataset on a single 3090 GPU.

| Number of samples | Training time (min)↓ | Mean(°)↓ | Med.(°)↓ | Acc@5°↑ |
|---|---|---|---|---|
| 0.6k | 122 | 5.8 | 2.8 | 0.80 |
| 4.6k | 122 | 5.3 | 2.6 | 0.82 |
| 37k | 136 | 5.3 | 2.6 | 0.83 |
| 295k | 168 | 5.3 | 2.5 | 0.82 |

Table 7: Per-category results ModelNet10-SO3 dataset.

| | | avg. | bathtub | bed | chair | desk | dresser | tv | n. stand | sofa | table | toilet |
|---|---|---|---|---|---|---|---|---|---|---|---|---|
| Acc@15°↑ | Deng et al. (2022) | 0.562 | 0.140 | 0.788 | 0.800 | 0.345 | 0.563 | 0.708 | 0.279 | 0.733 | 0.440 | 0.832 |
| | Prokudin et al. (2018) | 0.456 | 0.114 | 0.822 | 0.662 | 0.023 | 0.406 | 0.704 | 0.187 | 0.590 | 0.108 | 0.946 |
| | Mohlin et al. (2020) | 0.693 | 0.322 | 0.882 | 0.881 | 0.536 | 0.682 | 0.790 | 0.516 | 0.919 | 0.446 | 0.957 |
| | Murphy et al. (2021) | 0.719 | 0.392 | 0.877 | 0.874 | 0.615 | 0.687 | 0.799 | 0.567 | 0.914 | 0.523 | 0.945 |
| | Rotation Laplace | 0.741 | 0.390 | 0.902 | 0.909 | 0.644 | 0.722 | 0.815 | 0.590 | 0.934 | 0.521 | 0.977 |
| Acc@30°↑ | Deng et al. (2022) | 0.694 | 0.325 | 0.880 | 0.908 | 0.556 | 0.649 | 0.807 | 0.466 | 0.902 | 0.485 | 0.958 |
| | Prokudin et al. (2018) | 0.528 | 0.175 | 0.847 | 0.777 | 0.061 | 0.500 | 0.788 | 0.306 | 0.673 | 0.183 | 0.972 |
| | Mohlin et al. (2020) | 0.757 | 0.403 | 0.908 | 0.935 | 0.674 | 0.739 | 0.863 | 0.614 | 0.944 | 0.511 | 0.981 |
| | Murphy et al. (2021) | 0.735 | 0.410 | 0.883 | 0.917 | 0.629 | 0.688 | 0.832 | 0.570 | 0.921 | 0.531 | 0.967 |
| | Rotation Laplace | 0.770 | 0.430 | 0.911 | 0.940 | 0.698 | 0.751 | 0.869 | 0.625 | 0.946 | 0.541 | 0.986 |
| Median Error (°)↓ | Deng et al. (2022) | 32.6 | 147.8 | 9.2 | 8.3 | 25.0 | 11.9 | 9.8 | 36.9 | 10.0 | 58.6 | 8.5 |
| | Prokudin et al. (2018) | 49.3 | 122.8 | 3.6 | 9.6 | 117.2 | 29.9 | 6.7 | 73.0 | 10.4 | 115.5 | 4.1 |
| | Mohlin et al. (2020) | 17.1 | 89.1 | 4.4 | 5.2 | 13.0 | 6.3 | 5.8 | 13.5 | 4.0 | 25.8 | 4.0 |
| | Murphy et al. (2021) | 21.5 | 161.0 | 4.4 | 5.5 | 7.1 | 5.5 | 5.7 | 7.5 | 4.1 | 9.0 | 4.8 |
| | Rotation Laplace | 12.2 | 85.1 | 2.3 | 3.4 | 5.4 | 2.7 | 3.7 | 4.8 | 2.1 | 9.6 | 2.5 |

As stated in Table 6, the approximation with too few samples leads to inferior performance, and increasing the number of samples yields a better performance at the cost of a longer runtime. The performance improvement saturates when the number of samples is sufficient. We choose to use 37k samples in our experiments.

# E  ADDITIONAL RESULTS

## E.1  ADDITIONAL NUMERICAL RESULTS

Table 7 and 8 extend the results on ModelNet10-SO3 dataset and Pascal3D+ dataset in the main paper and show the per-category results. Our prediction with Rotation Laplace distribution is at or near state-of-the-art on many categories. The numbers for baselines are quoted from Murphy et al. (2021).

## E.2  ADDITIONAL VISUAL RESULTS

We show additional visual results on ModelNet10-SO3 dataset in Figure 5 and on Pascal3D+ dataset in Figure 6. As shown in the figures, our distribution provides rich information about the rotation estimations.

To visualize the predicted distributions, we adopt two popular visualization methods used in Mohlin et al. (2020) and Murphy et al. (2021). The visualization in Mohlin et al. (2020) is achieved by summing the three marginal distributions over the standard basis of $\mathbb{R}^3$ and displaying them on the sphere with color coding. Murphy et al. (2021) introduces a new visualization method based on discretization over SO(3). It projects a great circle of points on SO(3) to each point on the 2-sphere, and then uses the color wheel to indicate the location on the great circle. The probability density is shown by the size of the points on the plot. See the corresponding papers for more details.

Table 8: Per-category results on Pascal3D+ dataset.

| | | avg. | aero | bike | boat | bottle | bus | car | chair | table | mbike | sofa | train | tv |
|---|---|---|---|---|---|---|---|---|---|---|---|---|---|---|
| Acc@30°↑ | Tulsiani & Malik (2015) | 0.808 | 0.81 | 0.77 | 0.59 | 0.93 | 0.98 | 0.89 | 0.80 | 0.62 | 0.88 | 0.82 | 0.80 | 0.80 |
| | Mahendran et al. (2018) | 0.859 | 0.87 | 0.81 | 0.64 | 0.96 | 0.97 | 0.95 | 0.92 | 0.67 | 0.85 | 0.97 | 0.82 | 0.88 |
| | Liao et al. (2019) | 0.819 | 0.82 | 0.77 | 0.55 | 0.93 | 0.95 | 0.94 | 0.85 | 0.61 | 0.80 | 0.95 | 0.83 | 0.82 |
| | Prokudin et al. (2018) | 0.838 | 0.89 | 0.83 | 0.46 | 0.96 | 0.93 | 0.90 | 0.80 | 0.76 | 0.90 | 0.90 | 0.82 | 0.91 |
| | Mohlin et al. (2020) | 0.825 | 0.90 | 0.85 | 0.57 | 0.94 | 0.95 | 0.96 | 0.78 | 0.62 | 0.87 | 0.85 | 0.77 | 0.84 |
| | Murphy et al. (2021) | 0.837 | 0.81 | 0.85 | 0.56 | 0.93 | 0.95 | 0.94 | 0.87 | 0.78 | 0.85 | 0.88 | 0.78 | 0.86 |
| | Rot. Laplace (Ours) | 0.876 | 0.90 | 0.90 | 0.60 | 0.96 | 0.98 | 0.96 | 0.91 | 0.76 | 0.88 | 0.97 | 0.81 | 0.88 |
| Median error (°)↓ | Tulsiani & Malik (2015) | 13.6 | 13.8 | 17.7 | 21.3 | 12.9 | 5.8 | 9.1 | 14.8 | 15.2 | 14.7 | 13.7 | 8.7 | 15.4 |
| | Mahendran et al. (2018) | 10.1 | 8.5 | 14.8 | 20.5 | 7.0 | 3.1 | 5.1 | 9.3 | 11.3 | 14.2 | 10.2 | 5.6 | 11.7 |
| | Liao et al. (2019) | 13.0 | 13.0 | 16.4 | 29.1 | 10.3 | 4.8 | 6.8 | 11.6 | 12.0 | 17.1 | 12.3 | 8.6 | 14.3 |
| | Prokudin et al. (2018) | 12.2 | 9.7 | 15.5 | 45.6 | 5.4 | 2.9 | 4.5 | 13.1 | 12.6 | 11.8 | 9.1 | 4.3 | 12.0 |
| | Mohlin et al. (2020) | 11.5 | 10.1 | 15.6 | 24.3 | 7.8 | 3.3 | 5.3 | 13.5 | 12.5 | 12.9 | 13.8 | 7.4 | 11.7 |
| | Murphy et al. (2021) | 10.3 | 10.8 | 12.9 | 23.4 | 8.8 | 3.4 | 5.3 | 10.0 | 7.3 | 13.6 | 9.5 | 6.4 | 12.3 |
| | Rot. Laplace (Ours) | 9.4 | 8.6 | 11.7 | 21.8 | 6.9 | 2.8 | 4.8 | 7.9 | 9.1 | 12.2 | 8.1 | 6.9 | 11.6 |

| Input image | Distribution visual. (Mohlin et al., 2020) | Distribution visual. (Murphy et al., 2021) | Input image | Distribution visual. (Mohlin et al., 2020) | Distribution visual. (Murphy et al., 2021) |

Figure 5: **Visual results on ModelNet10-SO3 dataset.** We adopt the distribution visualization methods in Mohlin et al. (2020) and Murphy et al. (2021). For input images and visualizations with Mohlin et al. (2020), predicted rotations are shown with thick lines and the ground truths are with thin lines. For visualizations with Murphy et al. (2021), ground truths are shown by solid circles.

## F  DERIVATIONS

**Proposition 1 in the main paper.** *Let* $\boldsymbol{\Phi} = \log \widetilde{\mathbf{R}} \in \mathfrak{so}(3)$ *and* $\boldsymbol{\phi} = \boldsymbol{\Phi}^{\vee} \in \mathbb{R}^3$. *For rotation matrix* $\mathbf{R} \in \mathrm{SO}(3)$ *following* matrix Fisher distribution, *when* $\|\mathbf{R} - \mathbf{R}_0\| \to 0$ , $\boldsymbol{\phi}$ *follows zero-mean* multivariate Gaussian distribution.

*Proof.* For $\mathbf{R} \sim \mathcal{MF}(\mathbf{A})$, we have

$$p(\mathbf{R})\mathrm{d}\mathbf{R} \propto \exp\left(\mathrm{tr}(\mathbf{A}^{\mathrm{T}}\mathbf{R})\right)\mathrm{d}\mathbf{R} = \exp\left(\mathrm{tr}(\mathbf{S}\mathbf{V}^{\mathrm{T}}\widetilde{\mathbf{R}}\mathbf{V})\right)\mathrm{d}\widetilde{\mathbf{R}} \tag{9}$$

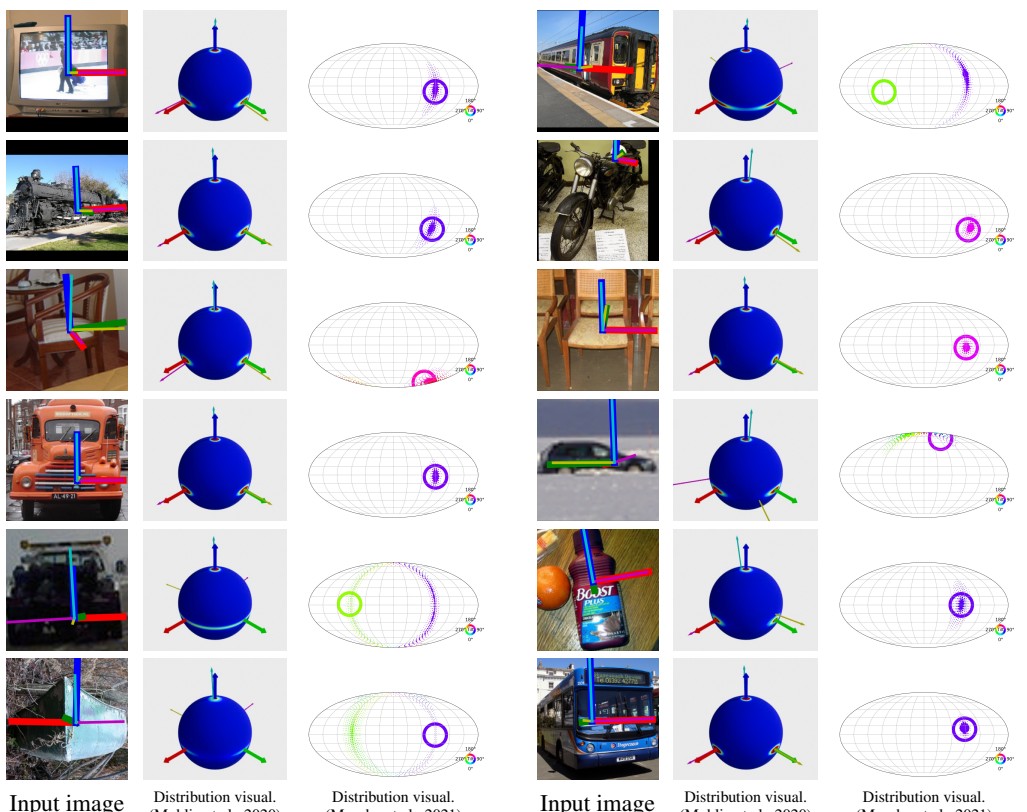

Figure 6: **Visual results on Pascal3D+ dataset.** We adopt the distribution visualization methods in Mohlin et al. (2020) and Murphy et al. (2021). For input images and visualizations with Mohlin et al. (2020), predicted rotations are shown with thick lines and the ground truths are with thin lines. For visualizations with Murphy et al. (2021), ground truths are shown by solid circles.

Considering Eq. 5 in the main paper, we have

$$
\begin{aligned}
\operatorname{tr}(\mathbf{S}\mathbf{V}^{\mathrm{T}}\widetilde{\mathbf{R}}\mathbf{V}) &= \operatorname{tr}(\mathbf{S}) + \sum_{(i,j,k)\in I} -\frac{1}{2}(s_j + s_k)\mu_i^2 + O(\|\boldsymbol{\phi}\|^3) \\
&= \operatorname{tr}(\mathbf{S}) - \frac{1}{2}\boldsymbol{\phi}^T\mathbf{V}\begin{bmatrix} s_2+s_3 & & \\ & s_1+s_3 & \\ & & s_1+s_2 \end{bmatrix}\mathbf{V}^T\boldsymbol{\phi}
\end{aligned}
\tag{10}
$$

Thus

$$
\begin{aligned}
p(\mathbf{R})\mathrm{d}\mathbf{R} &\propto \exp\left(\operatorname{tr}(\mathbf{A}^{\mathrm{T}}\mathbf{R})\right)\mathrm{d}\mathbf{R} \\
&= \frac{\exp(\operatorname{tr}(\mathbf{S}))}{8\pi^2}\exp\left(-\frac{1}{2}\boldsymbol{\phi}^T\boldsymbol{\Sigma}^{-1}\boldsymbol{\phi}\right)\left(1 + O(\|\boldsymbol{\phi}\|^2)\right)\mathrm{d}\boldsymbol{\phi}
\end{aligned}
\tag{11}
$$

When $\|\mathbf{R}-\mathbf{R}_0\| \to 0$, we have $\|\widetilde{\mathbf{R}}-\mathbf{I}\| \to 0$ and $\boldsymbol{\phi} \to \mathbf{0}$, so Eq. 11 follows the multivariate Gaussian distribution with the covariance matrix as $\boldsymbol{\Sigma}$, where $\boldsymbol{\Sigma} = \mathbf{V}\operatorname{diag}(\frac{1}{s_2+s_3}, \frac{1}{s_1+s_3}, \frac{1}{s_1+s_2})\mathbf{V}^T$. $\quad\square$

**Proposition 3 in the main paper.** *Denote $\mathbf{q}_0$ as the mode of Quaternion Laplace distribution. Let $\pi$ be the tangent space of $\mathbb{S}^3$ at $\mathbf{q}_0$, and $\pi(\mathbf{x}) \in \mathbb{R}^4$ be the projection of $\mathbf{x} \in \mathbb{R}^4$ on $\pi$. For quaternion $\mathbf{q} \in \mathbb{S}^3$ following Bingham distribution / Quaternion Laplace distribution, when $\mathbf{q} \to \mathbf{q}_0$, $\pi(\mathbf{q})$ follows zero-mean multivariate Gaussian distribution / zero-mean multivariate Laplace distribution.*

*Proof.* Denote $\mathbf{q_I} = (1,0,0,0)^T$ as the identity quaternion. Define $\mathbf{M}$ as an orthogonal matrix such that $\mathbf{M}^T\mathbf{q}_0 = \mathbf{q_I}$. Given $\pi(\mathbf{q}) = \mathbf{q} - (\mathbf{q} \cdot \mathbf{q}_0)\mathbf{q}_0$, we have

$$
\mathbf{M}^T\pi(\mathbf{q}) = \mathbf{M}^T\mathbf{q} - ((\mathbf{M}^T\mathbf{q}) \cdot (\mathbf{M}^T\mathbf{q}_0))\mathbf{q_I} = \mathbf{M}^T\mathbf{q} - w\mathbf{q_I},
\tag{12}
$$

where $\mathbf{M}^T\mathbf{q} = (w,x,y,z)^T$. Let $(\mathbf{e}_0, \mathbf{e}_1, \mathbf{e}_2, \mathbf{e}_3)$ be the column vectors of $\mathbf{I}_{4\times4}$, we have

$$
(\mathbf{M}\mathbf{e}_i) \cdot \mathbf{q}_0 = \mathbf{e}_i \cdot \mathbf{q_I} = 0
\tag{13}
$$

for $i = 1, 2, 3$. Therefore, $\mathbf{Me}_i (i = 1, 2, 3)$ form an orthogonal basis of $\pi$.

Given $\mathbf{M}^T \mathbf{q} = w\mathbf{e}_0 + x\mathbf{e}_1 + y\mathbf{e}_2 + z\mathbf{e}_3$, we have

$$\mathbf{q} = w(\mathbf{Me}_0) + x(\mathbf{Me}_1) + y(\mathbf{Me}_2) + z(\mathbf{Me}_3) \tag{14}$$

Therefore, $\boldsymbol{\eta} = (x, y, z)$ is the coordinate of $\pi(\mathbf{q})$ in $\pi$ under the basis of $\mathbf{Me}_i$.

The Jacobian of the transformation $\mathbf{q} \rightarrow \boldsymbol{\eta}$ is given by

$$
\begin{aligned}
\mathbf{J} = \frac{\partial \mathbf{q}}{\partial \boldsymbol{\eta}} &= \mathbf{M}\frac{\partial \left(\mathbf{M}^T \mathbf{q}\right)}{\partial \boldsymbol{\eta}} \\
&= \mathbf{M} \begin{bmatrix} -x/w & 1 & 0 & 0 \\ -y/w & 0 & 1 & 0 \\ -z/w & 0 & 0 & 0 \end{bmatrix}
\end{aligned}
\tag{15}
$$

Therefore, the scaling factor from $\boldsymbol{\eta}$ to $\mathbf{q}$ is given by

$$\frac{\mathrm{d}\mathbf{q}}{\mathrm{d}\boldsymbol{\eta}} = \det(\mathbf{J}\mathbf{J}^T) = 1 + \frac{x^2 + y^2 + z^2}{w^2} + O(\|\boldsymbol{\eta}\|^4) = 1 + O(\|\boldsymbol{\eta}\|^2). \tag{16}$$

Thus

$$
\begin{aligned}
\mathbf{q}^T \mathbf{MZM}^T \mathbf{q} &= \begin{bmatrix} w & x & y & z \end{bmatrix} \begin{bmatrix} 0 & & & \\ & z_1 & & \\ & & z_2 & \\ & & & z_3 \end{bmatrix} \begin{bmatrix} w \\ x \\ y \\ z \end{bmatrix} \\
&= \begin{bmatrix} x & y & z \end{bmatrix} \begin{bmatrix} z_1 & & \\ & z_2 & \\ & & z_3 \end{bmatrix} \begin{bmatrix} x \\ y \\ z \end{bmatrix} \\
&= \boldsymbol{\eta} \widetilde{\mathbf{Z}} \boldsymbol{\eta}
\end{aligned}
\tag{17}
$$

where we define $\widetilde{\mathbf{Z}} = \mathrm{diag}(z_1, z_2, z_3)$.

For Bingham distribution, we have

$$
\begin{aligned}
p(\mathbf{q})\mathrm{d}\mathbf{q} &\propto \exp\left(\mathbf{q}^T \mathbf{MZM}^T \mathbf{q}\right)\mathrm{d}\mathbf{q} \\
&= \exp\left(\boldsymbol{\eta}^T \widetilde{\mathbf{Z}} \boldsymbol{\eta}\right)(1 + O(\|\boldsymbol{\eta}\|^2))\mathrm{d}\boldsymbol{\eta} \\
&= \exp\left(-\boldsymbol{\eta}^T \boldsymbol{\Sigma}^{-1} \boldsymbol{\eta}\right)(1 + O(\|\boldsymbol{\eta}\|^2))\mathrm{d}\boldsymbol{\eta}
\end{aligned}
\tag{18}
$$

which follows the multivariate Gaussian distribution with the covariance matrix as $\boldsymbol{\Sigma}$, where $\boldsymbol{\Sigma} = -\mathrm{diag}(\frac{1}{z_1}, \frac{1}{z_2}, \frac{1}{z_3})$

For Quaternion Laplace distribution, we have

$$
\begin{aligned}
p(\mathbf{q})\mathrm{d}\mathbf{q} &\propto \frac{\exp\left(-\sqrt{-\mathbf{q}^T \mathbf{MZM}^T \mathbf{q}}\right)}{\sqrt{-\mathbf{q}^T \mathbf{MZM}^T \mathbf{q}}}\mathrm{d}\mathbf{q} \\
&= \frac{1}{\sqrt{2}}\frac{\exp\left(-\sqrt{-\boldsymbol{\eta}^T \widetilde{\mathbf{Z}} \boldsymbol{\eta}}\right)}{\sqrt{-\boldsymbol{\eta}^T \widetilde{\mathbf{Z}} \boldsymbol{\eta}}}(1 + O(\|\boldsymbol{\eta}\|^2))\mathrm{d}\boldsymbol{\eta} \\
&= \frac{1}{\sqrt{2}}\frac{\exp\left(-\sqrt{2\boldsymbol{\eta}^T \boldsymbol{\Sigma}^{-1} \boldsymbol{\eta}}\right)}{\sqrt{2\boldsymbol{\eta}^T \boldsymbol{\Sigma}^{-1} \boldsymbol{\eta}}}(1 + O(\|\boldsymbol{\eta}\|^2))\mathrm{d}\boldsymbol{\eta}
\end{aligned}
\tag{19}
$$

which follows the multivariate Laplace distribution with the covariance matrix as $\boldsymbol{\Sigma}$, where $\boldsymbol{\Sigma} = -2\mathrm{diag}(\frac{1}{z_1}, \frac{1}{z_2}, \frac{1}{z_3})$. □

**Proposition 4 in the main paper.** *Denote $\gamma$ as the standard transformation from unit quaternions to corresponding rotation matrices. For rotation matrix $\mathbf{R} \in \mathrm{SO}(3)$ following Rotation Laplace distribution, $\mathbf{q} = \gamma^{-1}(\mathbf{R}) \in \mathbb{S}^3$ follows Quaternion Laplace distribution.*

*Proof.* For a quaternion $\mathbf{q} = [q_0, q_1, q_2, q_3]$, we use the standard transform function $\gamma$ to compute its corresponding rotation matrix:

$$\gamma(\mathbf{q}) = \begin{bmatrix} 1 - 2q_2^2 - 2q_3^2 & 2q_1 q_2 - 2q_0 q_3 & 2q_1 q_3 + 2q_0 q_2 \\ 2q_1 q_2 + 2q_0 q_3 & 1 - 2q_1^2 - 2q_3^2 & 2q_2 q_3 - 2q_0 q_1 \\ 2q_1 q_3 - 2q_0 q_2 & 2q_2 q_3 + 2q_0 q_1 & 1 - 2q_1^2 - 2q_2^2 \end{bmatrix} \tag{20}$$

Let $\mathbf{u} = \gamma^{-1}(\mathbf{U}), \mathbf{v} = \gamma^{-1}(\mathbf{V})$ and

$$\widetilde{\mathbf{q}} = [\widetilde{q}_0, \widetilde{q}_1, \widetilde{q}_2, \widetilde{q}_3]^T = \gamma^{-1}\left(\mathbf{U}^T\mathbf{R}\mathbf{V}\right) = \bar{\mathbf{u}}\mathbf{q}\mathbf{v} \tag{21}$$

Note that the transformation $\mathbf{q} \rightarrow \bar{\mathbf{u}}\mathbf{q}\mathbf{v}$ is an orthogonal transformation on $\mathbb{S}^3$. Therefore, there exists an orthogonal Matrix $\mathbf{M}$, such that

$$\mathbf{M}^T\mathbf{q} = \bar{\mathbf{u}}\mathbf{q}\mathbf{v} = \widetilde{\mathbf{q}} \tag{22}$$

The scaling factor from quaternions to rotation matrices is given by

$$\mathrm{d}\mathbf{R} = \frac{1}{2\pi^2}\mathrm{d}\mathbf{q} \tag{23}$$

Suppose $\mathbf{R}$ follows Quaternion Laplace distribution as

$$p(\mathbf{R})\mathrm{d}\mathbf{R} = \frac{1}{F}\frac{\exp\left(-\sqrt{\mathrm{tr}(\mathbf{S}\text{-}\mathbf{A}^T\mathbf{R})}\right)}{\sqrt{\mathrm{tr}(\mathbf{S}\text{-}\mathbf{A}^T\mathbf{R})}}\mathrm{d}\mathbf{R} \tag{24}$$

Given

$$\mathrm{tr}(\mathbf{S}\text{-}\mathbf{A}^T\mathbf{R}) = \mathrm{tr}(\mathbf{S}\text{-}\mathbf{S}\mathbf{U}^T\mathbf{R}\mathbf{V}) = \sum_{(i,j,k)\in I} 2(s_j + s_k)q_i^2$$

$$= 2\widetilde{\mathbf{q}}^T \begin{bmatrix} 0 & & & \\ & s_2+s_3 & & \\ & & s_1+s_3 & \\ & & & s_1+s_2 \end{bmatrix} \widetilde{\mathbf{q}} \tag{25}$$

we have

$$
\begin{aligned}
p(\mathbf{R})\mathrm{d}\mathbf{R} &= \frac{1}{2\pi^2 F}\frac{\exp\left(-\sqrt{2\widetilde{\mathbf{q}}^T \begin{bmatrix} 0 & & & \\ & s_2+s_3 & & \\ & & s_1+s_3 & \\ & & & s_1+s_2 \end{bmatrix} \widetilde{\mathbf{q}}}\right)}{\sqrt{2\widetilde{\mathbf{q}}^T \begin{bmatrix} 0 & & & \\ & s_2+s_3 & & \\ & & s_1+s_3 & \\ & & & s_1+s_2 \end{bmatrix} \widetilde{\mathbf{q}}}}\mathrm{d}\mathbf{q} \\
&= \frac{1}{2\pi^2 F}\frac{\exp\left(-\sqrt{2\mathbf{q}^T\mathbf{M} \begin{bmatrix} 0 & & & \\ & s_2+s_3 & & \\ & & s_1+s_3 & \\ & & & s_1+s_2 \end{bmatrix} \mathbf{M}^T\mathbf{q}}\right)}{\sqrt{2\mathbf{q}^T\mathbf{M} \begin{bmatrix} 0 & & & \\ & s_2+s_3 & & \\ & & s_1+s_3 & \\ & & & s_1+s_2 \end{bmatrix} \mathbf{M}^T\mathbf{q}}}\mathrm{d}\mathbf{q} \\
&= \frac{1}{2\pi^2 F}\frac{\exp\left(-\sqrt{-\mathbf{q}^T\mathbf{M}\mathbf{Z}\mathbf{M}^T\mathbf{q}}\right)}{\sqrt{-\mathbf{q}^T\mathbf{M}\mathbf{Z}\mathbf{M}^T\mathbf{q}}}\mathrm{d}\mathbf{q},
\end{aligned} \tag{26}
$$

where $\mathbf{M}$ is an orthogonal matrix and $\mathbf{Z} = -2\,\mathrm{diag}(0, s_2 + s_3, s_1 + s_3, s_1 + s_2)$ is a $4 \times 4$ diagonal matrix. $\qquad\square$

**Elaboration of Eq. 3 in the main paper**

Given $\mathbf{R}_0 = \mathbf{U}\mathbf{V}^T$ and $\widetilde{\mathbf{R}} = \mathbf{R}_0^T\mathbf{R}$,

$$
\begin{aligned}
p(\mathbf{R})\mathrm{d}\mathbf{R} &\propto \frac{\exp\left(\sqrt{\mathrm{tr}(\mathbf{S} - \mathbf{A}^T\mathbf{R})}\right)}{\sqrt{\mathrm{tr}(\mathbf{S} - \mathbf{A}^T\mathbf{R})}}\mathrm{d}\mathbf{R} = \frac{\exp\left(\sqrt{\mathrm{tr}(\mathbf{S} - \mathbf{V}\mathbf{S}\mathbf{U}^T\mathbf{R})}\right)}{\sqrt{\mathrm{tr}(\mathbf{S} - \mathbf{V}\mathbf{S}\mathbf{U}^T\mathbf{R})}}\mathrm{d}\mathbf{R} = \frac{\exp\left(\sqrt{\mathrm{tr}(\mathbf{S} - \mathbf{S}\mathbf{U}^T\mathbf{R}\mathbf{V})}\right)}{\sqrt{\mathrm{tr}(\mathbf{S} - \mathbf{S}\mathbf{U}^T\mathbf{R}\mathbf{V})}}\mathrm{d}\mathbf{R} \\
&= \frac{\exp\left(\sqrt{\mathrm{tr}(\mathbf{S} - \mathbf{S}\mathbf{U}^T\mathbf{R}_0\widetilde{\mathbf{R}}\mathbf{V})}\right)}{\sqrt{\mathrm{tr}(\mathbf{S} - \mathbf{S}\mathbf{U}^T\mathbf{R}_0\widetilde{\mathbf{R}}\mathbf{V})}}\mathrm{d}\mathbf{R} = \frac{\exp\left(\sqrt{\mathrm{tr}(\mathbf{S} - \mathbf{S}\mathbf{V}^T\widetilde{\mathbf{R}}\mathbf{V})}\right)}{\sqrt{\mathrm{tr}(\mathbf{S} - \mathbf{S}\mathbf{V}^T\widetilde{\mathbf{R}}\mathbf{V})}}\mathrm{d}\mathbf{R}
\end{aligned} \tag{27}
$$

# G MORE IMPLEMENTATION DETAILS

For fair comparisons, we follow the implementation designs of Mohlin et al. (2020) and merely change the distribution from matrix Fisher distribution to our Rotation Laplace distribution. We use pretrained ResNet-101 as our backbone, and encode the object class information (for single-model-all-category experiments) by an embedding layer that produces a 32-dim vector. We apply a 512-512-9 MLP as the output layer.

The batch size is set as 32. We use the SGD optimizer and start with the learning rate of 0.01. For ModelNet10-SO3 dataset, we train 50 epochs with learning rate decaying by a factor of 10 at epochs 30, 40, and 45. For Pascal3D+ dataset, we train 120 epochs with the same learning rate decay at epochs 30, 60 and 90.

