# OpenReview forum: "A Laplace-inspired Distribution on SO(3) for Probabilistic Rotation Estimation"
_ICLR.cc/2023/Conference — ICLR 2023 notable top 25%_

### Official Review · Reviewer_biJ1 · 2022-10-17

**Confidence:** 4
**Correctness:** 4
**Technical Novelty And Significance:** 3
**Empirical Novelty And Significance:** 4
**Recommendation:** 8

**Clarity, Quality, Novelty And Reproducibility:**

The derivation of the Rotation Laplace distributions is somewhat straightforward, but clear, precise and original. One minor point is at the equation above (2), when you consider the 3D Laplace distribution. The density functions needs a clarification as to why it has collapsed into the fraction on the right hand side, as not many people are familiar with the modified Bessel function of the second kind. Another is that in definition 2 of section 4.1, the authors refer to the normalization factor as a constant F, when it actually depends on $\Sigma$.

I would encourage the authors to cite a classical paper about multivariate Laplace distribution rather than an evolving, unofficial Wikipedia page, one of which can be:

Kozubowski, Tomasz J.; Podgorski, Krzysztof; Rychlik, Igor (2010). "Multivariate Generalize Laplace Distributions and Related Random Fields" (PDF). University of Gothenburg. Retrieved 2017-05-28.

Other than the missing pieces in section 4.3, I think the remaining materials are enough for proper reproduction of the results.

**Strength And Weaknesses:**

### Strength

The motivation of the paper is clear. The results convincingly support the motivation. In Euclidean spaces, Laplace distributions, while less studied, have proven to be more robust than Gaussian distributions.  This paper leverages that idea from Euclidean spaces to SO(3). In this context, I find figure 2 very useful and straight to the point.

### Weaknesses

The section 4.3 that describes dense discrete approximation of the normalizing factor needs further elaborating. For example, given that this part is key for forward and backward passes, it would be critical to know how many points have been used to quantise SO(3) via the Hopf fibration. What kind of speed-accuracy tradeoff have you experienced? Obviously, if we go to rotation spaces higher than 3D, this quantisation approach is unlikely to work.

Given that the contribution is a distribution family, a rather general concept, it would be good to have applications other than just robust rotation regression.

**Summary Of The Paper:**

The paper introduces a new family of probability distributions on SO(3) called Rotation Laplace distributions. The literature has established how a matrix Fisher distribution on SO(3) approximates a zero-mean multivariate Gaussian distribution on the tangent space $T_{\mathbf{R}_0}SO(3)$, where $\mathbf{R}_0$ is the mode rotation of the distribution. Using exactly the same way, the paper shows how a Rotation Laplace distribution approximates a zero-mean multivariate Laplace distribution on the tangent space at the mode rotation.

The paper does not stop there. As each matrix Fisher distribution on SO(3) has an equivalent Bingham distribution on the unit quaternion space, a family of Quaternion Laplace distributions is introduced, members of which, proved in the same way, are equivalent to Rotation Laplace distributions.

Just like matrix Fisher distributions, the proposed distributions suffer from having no analytical form for the normalization factor. In the paper, the authors relied on quantising SO(3) using Hopf fibration and pre-computing or approximating the normalization factor and their first-order derivatives at quantised points, in the same way as state-of-the-art approaches.

Experiments on the ModelNet10-SO3 dataset and the Pascal3D+ dataset show results favouring the proposed distributions, over existing distributions on and representations of rotation matrices, in the problem of robust rotation regression.



**Summary Of The Review:**

Overall, I find the paper a good read. A clear motivation supported by good results, despite that the technical contribution is rather straight forward to derive.

---

> ### Author Response · Authors · 2022-11-13
> **Author Response**
>
> We thank the reviewer for the valuable feedback. Here are our responses.
>
> (Changes in the main paper and supplementary are highlighted in red. The supplementary has been moved to the end of the main paper.)
>
> ## Computation of the normalization factors
>
> The normalization factors can be computed through discretization over SO(3). For computational efficiency, as mentioned in the last paragraph of Section 4.3 of the main paper, we further build a lookup table w.r.t. the proper singular values of the distribution parameter for both forward and backward pass, where each element in the table is the corresponding factor and gradient computed through discretization. We then apply trilinear interpolation to obtain the factor and gradient for the query singular values. The use of lookup tables avoids the online calculation and makes the process fast (~1ms for forward and backward, respectively). This strategy is also used in [1, 2] and we do not claim contributions on it.
>
> For computing each element in the table, we discretize SO(3) space into about 300k equivolumetric grids. We sample about 5.6M different sets of proper singular values, the storage for which is about 21MB for forward and 63MB for backward. To validate the effectiveness of the strategy, we conduct an ablation study on ModelNet10-SO3 toilet dataset with either online computation (with 300k samples) or lookup tables, and the results are similar (Mean: 5.3 vs 5.3, Median: 2.5 vs 2.6, Acc5: 0.82 vs 0.83).
>
> We have revised Section 4.3 of the main paper for clearer exposition and incorporated more details in the supplementary.
>
> ## Discussion on our contributions
>
> The main focus of this work is to improve rotation estimation, other than generally discussing probability distributions on SO(3). We believe our contributions already deserve a clear accept, based on the following reasons.
>
> 1. Rotation regression itself is a sufficiently important research problem and has wide applications in computer vision, graphics, robotics and a broad area in AI for science. Due to its importance, recent years have witnessed more and more papers published in top conferences that only focus on improving rotation regression, including:
>
>    - Rotation representations: 6D (CVPR 2019 [3]), 9D (NeurIPS 2020 [4]), 10D (RSS 2020 [5])
>
>    - Rotation gradients: RPMG (CVPR 2022 [6])
>
>    - Rotation distributions: von Mises distributions (ECCV 2018 [7]), Bingham distribution (ICLR 2019 [1]) (IJCV 2022 [2]), matrix Fisher distribution (NeurIPS 2020 [8]), IPDF (ICML 2021 [9]).
>
>     Our work thus focuses only on tackling this problem.
>
> 2. Our work makes innovative contributions by proposing the Rotation Laplace distribution and advances this important field by achieving new state-of-the-art performance and significant improvements on all the benchmarks. It is expected that this work would benefit diverse downstream applications and generate broader impacts.
>
> 3. It would be unfair to give us a marginal accept if the reason is that we lack demonstrations of our newly proposed distributions in other fields beyond rotation regression. Works like [1, 2] on Bingham distribution and [8] on matrix Fisher distribution introduced the existing distributions (proposed back in the 1970s) into the field of deep rotation regression and all get accepted by top conferences. We believe novel works with new ideas/concepts in mathematics and statistics should be at least judged by the same criteria as works without contributions of new distributions, otherwise it in fact discourages people to do so. We thus argue that we should be able to safely leave demonstrating the other applications of our proposed distribution to future works.
>
> We sincerely wish the reviewer to rethink our contribution and we are willing to respond to your further questions.

---

> > ### Comment · Reviewer_biJ1 · 2022-11-22
> > **Thank you.**
> >
> > Thank you for this part of your response and for having updated the paper with new materials. The new materials have addressed my concerns.
> >
> > ### Regarding point 3 in the discussion on your contributions: "It would be unfair to give us a marginal accept if the reason is that we lack demonstrations of our newly proposed distributions in other fields beyond rotation regression."
> > In the original review, I did not give a marginal accept due to lack of demonstrations in other fields. In fact, the lack of demonstrations in other fields was only a minor point. The main points were:
> > 1. The paper is an integration paper. It integrates two ideas: switching from Gaussian to Laplacian for more robustness, and lifting the distributions from Euclidean to SO(3). Both ideas are very interesting. However, they are known in the literature. I could not give credit to those ideas when they are presented in this paper. Instead, I could only give credit to how the ideas were used and how the integration was done in this paper. On that front, however, I found that the integration was rather straight forward. There is a technical difficulty in computing the normalization factor, as it is often the most difficult part when inventing a new distribution family. But the solution was sampling with Hopf fibration. This is not bad. It says that the problem itself is hard but at the same time I could not give more credit to the paper's novelty.
> > 2. Lack of description of how Hopf fibration was conducted in the first draft. For me, this is crucial to the efficiency of the proposed approach. Fortunately, you have addressed it in the response.
> >
> > I have seen the new materials in the current draft. In my view, they make the paper a lot more solid. As a result, I have:
> > - increased my Empirical Novelty and Significance rating from 3 to 4.
> > - increased my overall rating from 6 to 8.

---

> > > ### Author Response · Authors · 2022-11-23
> > > **Thank you.**
> > >
> > > Dear Reviewer biJ1,
> > >
> > > Thanks for your response.
> > >
> > >
> > > Best,
> > >
> > > The authors

---

> ### Author Response · Authors · 2022-11-13
> **[Cont.] Author Response**
>
> ## Clarity
>
> - >*The density functions needs a clarification as to why it has collapsed into the fraction on the right hand side, as not many people are familiar with the modified Bessel function of the second kind.*
>
>   Thanks for the comment. We have revised the main paper. It is because of the property of the modified Bessel function of the second kind: $K_{-\frac{1}{2}}(\xi)\propto \xi^{-\frac{1}{2}} \exp (-\xi)$.
>
> - > *In definition 2 of section 4.1, the authors refer to the normalization factor as a constant F, when it actually depends on $\Sigma$*.
>
>   Thanks for pointing it out. We have revised the term "normalization constant" to "normalization factor" in the paper.
>
> - > *I would encourage the authors to cite a classical paper about multivariate Laplace distribution rather than an evolving, unofficial Wikipedia page*.
>
>   Thanks for the suggestion. We have replaced the reference.
>
> [1] Igor Gilitschenski, Roshni Sahoo, Wilko Schwarting, Alexander Amini, Sertac Karaman, and Daniela Rus. Deep orientation uncertainty learning based on a bingham loss. In International Conference on Learning Representations, 2019.
>
> [2] Haowen Deng, Mai Bui, Nassir Navab, Leonidas Guibas, Slobodan Ilic, and Tolga Birdal. Deep bingham networks: Dealing with uncertainty and ambiguity in pose estimation. International Journal of Computer Vision, pp. 1–28, 2022.
>
> [3] Yi Zhou, Connelly Barnes, Jingwan Lu, Jimei Yang, and Hao Li. On the continuity of rotation representations in neural networks. In Proceedings of the IEEE/CVF Conference on Computer Vision and Pattern Recognition, pp. 5745–5753, 2019.
>
> [4] Jake Levinson, Carlos Esteves, Kefan Chen, Noah Snavely, Angjoo Kanazawa, Afshin Rostamizadeh, and Ameesh Makadia. An analysis of svd for deep rotation estimation. Advances in Neural Information Processing Systems, 33:22554–22565, 2020.
>
> [5] Valentin Peretroukhin, Matthew Giamou, David M. Rosen, W. Nicholas Greene, Nicholas Roy, and Jonathan Kelly. A Smooth Representation of SO(3) for Deep Rotation Learning with Uncertainty. In Proceedings of Robotics: Science and Systems (RSS’20), Jul. 12–16 2020.
>
> [6] Jiayi Chen, Yingda Yin, Tolga Birdal, Baoquan Chen, Leonidas J Guibas, and He Wang. Projective manifold gradient layer for deep rotation regression. In Proceedings of the IEEE/CVF Conference on Computer Vision and Pattern Recognition, pp. 6646–6655, 2022.
>
> [7] Sergey Prokudin, Peter Gehler, and Sebastian Nowozin. Deep directional statistics: Pose estimation with uncertainty quantification. In Proceedings of the European conference on computer vision (ECCV), pp. 534–551, 2018.
>
> [8] David Mohlin, Josephine Sullivan, and G´erald Bianchi. Probabilistic orientation estimation with matrix fisher distributions. Advances in Neural Information Processing Systems, 33:4884–4893, 2020.
>
> [9] Kieran A Murphy, Carlos Esteves, Varun Jampani, Srikumar Ramalingam, and Ameesh Makadia. Implicit-pdf: Non-parametric representation of probability distributions on the rotation manifold. In International Conference on Machine Learning, pp. 7882–7893. PMLR, 2021

---

> > ### Comment · Reviewer_biJ1 · 2022-11-22
> > **Thank you.**
> >
> > Thank you for this part of your response. Much appreciated.

---

### Official Review · Reviewer_z3mk · 2022-10-24

**Confidence:** 4
**Correctness:** 2
**Technical Novelty And Significance:** 4
**Empirical Novelty And Significance:** 3
**Recommendation:** 6

**Clarity, Quality, Novelty And Reproducibility:**

Clarity:

Overall, the manuscript is clearly written yet there is a cruel lack of figures to guide the reader.

The maths are badly written with multiple undefined notation : so(3) ? exponent v ? R -> R_0 ? Any reference for proposition 1 ? In def 2 is d=k ? In equation (3), is it a U instead of V ? What is \hat phi ? How do you have such a parametrization ? Any ref ? Define the Haar measure ? In section 4.3 and 4.4 the equations giving the method to approximate the normalizing constant are useless : int becomes sum and d becomes \Delta. What are the differences ? What does these notations mean ?

In which way section 5.5 is an ablation study ?

There are references to the supplementary material but there is no supplementary material...


Quality:

The authors should refer to an academic reference when talking about multivariate Laplace distribution:
Eltoft, T., Kim, T., & Lee, T. W. (2006). On the multivariate Laplace distribution. IEEE Signal Processing Letters, 13(5), 300-303

The badly written theoretical sections make me doubt of the numerical sections.

Qualitative results would improve the result sections.


Novelty: I haven't seen such a distribution on SO(3) proposed before.


Reproducibility: There is no claim about whether the code will be added online.





**Strength And Weaknesses:**

Strength:
- extended numerical experiments showing the superiority of their estimation compared to others
- a proof that the rotation Laplace distribution is related to the multivariate Laplace distribution
- an extension to Quaternions

Weaknesses:
- the problem is not well introduced (what rotation are the authors trying to estimate ?)
- there are multiple undefined mathematical notations which makes the claims unverifiable or the equations useless
- no qualitative illustration of the results
- not enough figures to guide the reader

**Summary Of The Paper:**

The authors proposes a distribution on SO(3) to estimate 3 degrees of freedom rotations in RGB images. The distribution is inspired from the multi-dimensional Laplace distribution which will allow to have estimates that are robust to outliers. The authors also provide an equivalence between the matrix distribution on SO(3) and a vector distribution on the Quaternion. The method is extensively tested on multiple dataset of rotation and pose estimations.

**Summary Of The Review:**

To me, the current submission has two many flaws to be considered for acceptance.
Though, I will reconsider my opinion after revision.

---

> ### Author Response · Authors · 2022-11-06
> **An immediate response: About supplementary, clarification and reproducibility**
>
> We thank the reviewer for the comprehensive feedback and feel sorry for the unpleasant read. Due to possible misunderstandings, we feel the necessity to give an immediate response to some of the concerns and are more than happy to have further discussions.
>
> Please note that the authors are still working on resolving the remaining issues that are not mentioned in this response.
>
> We earnestly and eagerly request the reviewer to re-evaluate our work based on the clarification.
>
> ## Supplementary material
>
> We kindly note that the supplementary material is and has been in the system since the submission deadline (Sep 28th). Please find it (a zip file containing a single PDF file) in the "Supplementary Material" section of the OpenReview system. Some of the concerns have already been clarified:
>
> - Definition of $SO(3)$ and $\mathfrak{so}(3)$.
> - Definition of *vee* $\vee$ map and *hat* $\wedge$ map (mentioned as "exponent v" and "\hat phi" by the reviewer). This paper follows the common notations in Lie algebra and exponential \& logarithm map, and we also add some references.
> - The proofs of prop. 1, prop. 3 and prop. 4.
>
> ## Further clarification
>
> (Changes in the main paper and supplementary are highlighted in red)
>
> - > *What rotation are the authors trying to estimate?*
>
>   We estimate the object rotations from single RGB images, which is the same task in the previous state-of-the-arts (IPDF [1], matrix Fisher distribution [2], etc.). Object rotation is the relative rotation between the input object and the object in the canonical pose. We have added the descriptions in Sec. 5 of the main paper.
>
> - > *R -> R0?*
>
>   We use $\mathbf{R}\rightarrow \mathbf{R}_0$ to denote "$\mathbf{R}$ approaches $\mathbf{R}_0$". For preciseness, we replace it with $\\|\mathbf{R}-\mathbf{R_0}\\|\rightarrow 0$. We have revised the notations in both main paper and supplementary.
>
> - > *Any reference for proposition 1?*
>
>   We have added the references in Sec 3.2. Also, the proof is shown in the supplementary.
>
> - > *In def 2 is d=k?*
>
>   Thanks for pointing out this typo. We have revised the main paper to replace $k$ with $d$.
>
> - > *In equation (3), is it a U instead of V?*
>
>   No, it should be $\mathbf{V}$ as it is. We have added the elaboration in the supplementary and also show it below.
>   Given $\mathbf{R}_0 = \mathbf{UV}^T$ and $\mathbf{\widetilde{R}}=\mathbf{R}_0^T\mathbf{R}$,
>   $$
>   \small
>       p(\mathbf{R})\mathrm{d}\mathbf{R} \propto
>       \frac{\exp\left(\sqrt{\operatorname{tr}(\mathbf{S}-{\mathbf{A}^T\mathbf{R}})}\right)}{\sqrt{\operatorname{tr}(\mathbf{S}-{\mathbf{A}^T\mathbf{R}})}} \mathrm{d}\mathbf{R}
>       = \frac{\exp\left(\sqrt{\operatorname{tr}(\mathbf{S}-\mathbf{V}\mathbf{S}\mathbf{U}^T\mathbf{R})}\right)}{\sqrt{\operatorname{tr}(\mathbf{S}-\mathbf{V}\mathbf{S}\mathbf{U}^T\mathbf{R})}} \mathrm{d}\mathbf{R}
>       = \frac{\exp\left(\sqrt{\operatorname{tr}(\mathbf{S}-\mathbf{S}\mathbf{U}^T\mathbf{R}\mathbf{V})}\right)}{\sqrt{\operatorname{tr}(\mathbf{S}-\mathbf{S}\mathbf{U}^T\mathbf{R}\mathbf{V})}} \mathrm{d}\mathbf{R}
>       = \frac{\exp\left(\sqrt{\operatorname{tr}(\mathbf{S}-\mathbf{S}\mathbf{U}^T\mathbf{R}_0\mathbf{\widetilde{R}}\mathbf{V})}\right)}{\sqrt{\operatorname{tr}(\mathbf{S}-\mathbf{S}\mathbf{U}^T\mathbf{R}_0\mathbf{\widetilde{R}}\mathbf{V})}} \mathrm{d}\mathbf{R}
>       = \frac{\exp\left(\sqrt{\operatorname{tr}(\mathbf{S}-\mathbf{S}\mathbf{V}^T\mathbf{\widetilde{R}}\mathbf{V})}\right)}{\sqrt{\operatorname{tr}(\mathbf{S}-\mathbf{S}\mathbf{V}^T\mathbf{\widetilde{R}}\mathbf{V})}} \mathrm{d}\mathbf{R}
>   $$
>
> - > *Define the Haar measure?*
>
>   To evaluate the normalization factors and therefore the probability density functions, the measure $\mathrm{d}\mathbf{R}$ on $SO(3)$ needs to be defined. For the Lie group $SO(3)$, the commonly used bi-invariant measure is referred to as Haar measure [3,4]. Haar measure is unique up to scalar multiples [5] and we follow the common practice [2,6] that the Haar measure $\mathrm{d}\mathbf{R}$ is scaled such that $\int_{SO(3)} \mathrm{d} \mathbf{R}=1$. We have added the definition in the supplementary.

---

> ### Author Response · Authors · 2022-11-06
> **[Cont.] An immediate response: About supplementary, clarification and reproducibility**
>
> - > *What is the effect of equations in section 4.3 and 4.4 for the method to approximate the normalizing factors?*
>
>   Due to the unavailability of an analytical form for normalization factors, we approximate the factors by quantizing $SO(3)$. The $\int$ term represents the ideal (but not applicable) normalization factors, and the $\Sigma$ term is the approximated (and applicable) factors after discretization. A $\approx$ sign indicates the approximation. Denote the finite set of equivolumetric grids as $\mathcal{G}=\\{\mathbf{R}|\mathbf{R}\in SO(3)\\}$ and $\mathcal{G}_\mathbf{q}=\\{ \mathbf{q} | \mathbf{q}\in \mathcal{S}^3 \\}$,
>
>   then $\Delta \mathbf{R}=
>   \frac{\int_{SO(3)} \mathrm{d} \mathbf{R}}{|\mathcal{G}|}=\frac{1}{|\mathcal{G}|}$ and $\Delta \mathbf{q}=\frac{\oint_{\mathcal{S}^3} \mathrm{d}\mathbf{q}}{|\mathcal{G}_\mathbf{q}|} = \frac{2\pi^2}{|\mathcal{G}_\mathbf{q}|}$. We have revised the two sections for better exposition.
>
> - > *In which way section 5.5 is an ablation study?*
>
>   Sorry for the confusion. We tried to validate our choice to leverage the continuous rotation matrix representation. For better clarification, we have changed the title to "Comparisons of Rotation Laplace distribution and Quaternion Laplace distribution"
>
> - > *Reference of the multivariate Laplace distribution*
>
>   We have replaced the reference. Thanks for the suggestion.
>
> ## Reproducibility and reliability of results
>
> The authors will release the code under acceptance to guarantee the reliability of the numerical results.
>
> As mentioned in Sec 5.4, for a fair comparison, our implementation mostly follows [[2]](https://github.com/Davmo049/Public_prob_orientation_estimation_with_matrix_fisher_distributions) and the only difference is the parametrization of distributions.
>
>
>
> [1] Kieran A Murphy, Carlos Esteves, Varun Jampani, Srikumar Ramalingam, and Ameesh Makadia. Implicit-pdf: Non-parametric representation of probability distributions on the rotation manifold. In International Conference on Machine Learning, pp. 7882–7893. PMLR, 2021
>
> [2] David Mohlin, Josephine Sullivan, and G ́erald Bianchi. Probabilistic orientation estimation with matrix fisher distributions. Advances in Neural Information Processing Systems, 33:4884–4893, 2020
>
> [3] Alfred Haar. Der massbegriff in der theorie der kontinuierlichen gruppen. Annals of mathematics, pp. 147–169, 1933.
>
> [4] Ioan Mackenzie James. History of topology. Elsevier, 1999.
>
> [5] Gregory S Chirikjian. Engineering applications of noncommutative harmonic analysis: with emphasis on rotation and motion groups. CRC press, 2000.
>
> [6] Taeyoung Lee. Bayesian attitude estimation with the matrix fisher distribution on so (3). IEEE Transactions on Automatic Control, 63(10):3377–3392, 2018

---

> ### Author Response · Authors · 2022-11-13
> **Additional Response**
>
> (Changes in the main paper and supplementary are highlighted in red. The supplementary has been moved to the end of the main paper.)
>
> ## Qualitative results and analysis
>
> We have added the visualizations of predicted distributions in Section 5.4 (Figure 2) of the main paper, and more qualitative results are shown in Section E.2 (Figure 6 & 7) of the supplementary.
>
> As shown in Figure 2 of the main paper, we found the predicted distributions can exhibit high uncertainty when the object has rotational symmetry, leading to near 180$^\circ$ errors (subfigure a-c), or the input image is with low resolution (subfigure d). Subfigure (e-f) show cases with high certainty and reasonably low errors.
>
> Please see Section 5.4 of the main paper and Section E.2 of the supplementary for the visualizations.
>
> ## More analysis and figures
>
> In Section B of the supplementary, we add more analysis on gradient w.r.t. outliers, where Figure 3 of supplementary illustrates the gradient magnitudes $\\|\partial\mathcal{L}/\partial\text{(distribution param.)}\\|$ w.r.t. the prediction errors after convergence. As shown in the figure, for our Rotation Laplace distribution, the gradient magnitudes are not affected by the prediction errors much, leading to stable learning with robustness to outliers.
>
> In Section C of the supplementary, we conduct experiments by manually injecting outliers into the perfectly labeled synthetic dataset. As shown in Figure 4 of the supplementary, our method clearly better tolerates outliers than matrix Fisher distribution, resulting in less performance degradation and remains a reasonable performance even under intense perturbations, which shows the superior robustness of our method.
>
> In Section D of the supplementary, we quantify the prediction uncertainty in the form of distribution entropy. As shown in Figure 5 of the supplementary, predictions with lower entropies (i.e., lower uncertainty) clearly achieve higher accuracy than predictions with large entropies, demonstrating the ability of uncertainty estimation of our Rotation Laplace distribution.
>
> ## Looking forward to your reply
>
> The authors are eagerly looking forward to your reply and are willing to answer further questions.

---

> ### Author Response · Authors · 2022-11-26
> **Looking forward to your reply**
>
> Dear Reviewer z3mk,
>
> We eagerly look forward to hearing from you and answering further questions.
>
> Many thanks,
>
> The authors

---

### Official Review · Reviewer_eVhd · 2022-10-25

**Confidence:** 5
**Correctness:** 3
**Technical Novelty And Significance:** 4
**Empirical Novelty And Significance:** 4
**Recommendation:** 8

**Clarity, Quality, Novelty And Reproducibility:**

The paper is well written, clear presented.
The proposed distribution for SO(3) is novel.
The work should be able to reproduce given released code.

**Strength And Weaknesses:**

+ The proposed rotation Laplace distribution is novel, and the formulation is rigorous.
+ The paper is well written, with clear motivation, literature review, and methodology.
+ The results are persuasive, both analytically (Figure 1) and quantitatively.

- Despite the results are impressive, experiments do not sufficiently demonstrate the proposed regression framework is able to properly handle outliers, as it is claimed in the introduction. In other words, where do improvements come from is not clear nor persuasive. Probably theoretical/empirical analysis of gradient behavior w.r.t. outlier is missing.

- Regardless of this work is probabilistic rotation estimation, the presented results do not show the uncertainty quantification, nor demonstrate certain benefits in handling hard cases with rotational ambiguity (due to symmetry or lack of discriminate visual features). Adding certain analysis or showcases would be beneficial.


**Summary Of The Paper:**

The paper studies 3D rotation regression with probabilistic models.
A rotation Laplace distribution is proposed to place on SO(3), offering robustness in handling outliers in training deep regression neural networks. This novel distribution is properly defined, and the paper also outlines a practical approximation to implement with deep neural network. Results show a solid improvement over previous arts.

**Summary Of The Review:**


This paper is definitely a well written paper, which presents an interesting method to tackle probabilistic 3D rotation regression.
The empirical results show a solid improvement over other state-of-the-art methods. The reviewer feels pleasant to read the entire paper.
Despite the impressive results, the motivation of the paper lacks of theoretical and empirical support. In terms of sensitivity to outliers/imperfect annotations, one way to demonstrate the superior robustness of the proposed model over other methods could be injecting noise to perfectly labeled synthetic data. Besides, it is a pity that this paper does not show any uncertainty quantification of the proposed probabilistic model in estimating rotation of hard/ambiguous objects.

---

> ### Author Response · Authors · 2022-11-13
> **Author Response**
>
> We thank the reviewer for the fruitful suggestions and feedback. Here are our responses.
>
> (Changes in the main paper and supplementary are highlighted in red. The supplementary has been moved to the end of the main paper.)
>
> ## Experiments on ModelNet10-SO3 dataset with outlier injections
>
> As suggested, we conduct experiments on ModelNet10-SO3 dataset with outlier injections. Specifically, we randomly choose 1%, 5%, 10% and 30% images from the training set respectively, and apply a random rotation in SO(3) to the given ground truth.  Thus, the chosen images become outliers in the dataset due to perturbed annotations. We fix the processed dataset for different methods.
>
> The results are shown in Table 6 and Figure 4 of the supplementary, where our method consistently outperforms matrix Fisher distribution under different levels of perturbations. More importantly, as shown in Figure 4, our method clearly better tolerates the outliers, resulting in less performance degradation and remains a reasonable performance even under intense perturbations. For example, Acc@30$^\circ$ of matrix Fisher distribution greatly drops from 0.751 to 0.467 with 30% outliers, while that of our method merely goes down from 0.770 to 0.700, which shows the superior robustness of our method.
>
> Please see Section C of the supplementary for more details.
>
> ## More analysis on gradient w.r.t. outliers
>
> In Figure 3 of the supplementary, for matrix Fisher distribution and Rotation Laplace distribution, we visualize the gradient magnitudes $\\|\partial\mathcal{L}/\partial\text{(distribution param.)}\\|$ w.r.t. the prediction errors on ModelNet10-SO3 dataset after convergence, where each point is a data point in the test set. As shown in the figure, for matrix Fisher distribution, predictions with larger errors clearly yield larger gradient magnitudes, and those with near 180$^\circ$ errors (the outliers) have the biggest impact. Given that outliers may be inevitable and hard to be fixed, they may severely disturb the training process and the sensitivity to outliers can result in a poor fit [1, 2]. In contrast, for our Rotation Laplace distribution, the gradient magnitudes are not affected by the prediction errors much, leading to a stable learning process.
>
> Consistent results can also be seen in Figure 1 of the main paper, where the red dots illustrate the *sum* of the gradient magnitude over the population within an interval of prediction errors. We argue that, at convergence, the gradient should focus more on the large population with low errors rather than fixing the unavoidable large errors.
>
> We have incorporated this answer into Section B of the supplementary.
>
> ## Uncertainty quantification measured by distribution entropy
>
> Similar to [3], we quantify the prediction uncertainty in the form of distribution entropy. As shown in Figure 5 of the supplementary, predictions with lower entropies (i.e., lower uncertainty) clearly achieve higher accuracy than predictions with large entropies, demonstrating the ability of uncertainty estimation of our Rotation Laplace distribution.
>
> Please refer to Section D of the supplementary for more details.
>
> ## Qualitative results and analysis
>
> We have added the visualizations of predicted distributions in Section 5.4 (Figure 2) of the main paper, and more qualitative results are shown in Section E.2 (Figure 6 & 7) of the supplementary.
>
> As shown in Figure 2 of the main paper, we found the predicted distributions can exhibit high uncertainty when the object has rotational symmetry, leading to near 180$^\circ$ errors (subfigure a-c), or the input image is with low resolution (subfigure d). Subfigure (e-f) show cases with high certainty and reasonably low errors.
>
> Please see Section 5.4 of the main paper and Section E.2 of the supplementary for the visualizations.
>
> [1] Kevin P Murphy. Machine learning: a probabilistic perspective. MIT press, 2012.
>
> [2] Deebul S Nair, Nico Hochgeschwender, and Miguel A Olivares-Mendez. Maximum likelihood uncertainty estimation: Robustness to outliers. The Thirty-Sixth AAAI Conference on Artificial Intelligence (AAAI-22), 2022.
>
> [3] Yingda Yin, Yingcheng Cai, He Wang, and Baoquan Chen. Fishermatch: Semi-supervised rotation regression via entropy-based filtering. In Proceedings of the IEEE/CVF Conference on Computer Vision and Pattern Recognition, pp. 11164–11173, 2022.

---

### Author Response · Authors · 2022-11-13
**Supplementary has been moved to the end of the main paper**

For ease of reading, we have moved the supplementary to the end of the main paper. Previously, it was a separate file in the "Supplementary Material" section of the OpenReview system.

---

### Decision · Program_Chairs · 2023-01-20

**Decision:**

Accept: notable-top-25%

**Justification For Why Not Higher Score:**

Rotation regression is an important topic in particular computer vision. In this paper, the authors propose an interesting approach. Moreover, the most of reviewers puts high score. However, the potential audience of rotation regression is limited compared to a general deep learning contribution. So, it would be great to be presented as spotlight.

**Justification For Why Not Lower Score:**

Most of the reviewers are happy to accept; it should be presented as Accept (oral) or Accept (spotlight).

**Metareview: Summary, Strengths And Weaknesses:**

In this paper, the authors propose a rotation Laplace distribution, which offers robustness in handling outliers, for 3D rotation regression with probabilistic models. Then, the proposed rotation Laplace distribution is used for 3D rotation regression. The idea is novel and interesting. Overall, the reviewers agree to accept the paper; I also vote for acceptance.

In the camera-ready version, please address the issues raised by reviewers.

**Note From Pc:**

if the above contains the word "oral" or "spotlight" please see: "oral" presentation means -> notable-top-5% and "spotlight" means -> notable-top-25%. As stated in our emails, we are disassociating presentation type from AC recommendations